# Diagnostic Assessment and Restoration Plan for Damaged Forest around the Seokpo Zinc Smelter, Central Eastern Korea

A Reum Kim [1] , Bong Soon Lim [1] , Jaewon Seol [1], Chi Hong Lim [2] , Young Han You [3], Wan Sup Lee [4] and Chang Seok Lee [1],*

1   Division of Chemistry and Bio-Environmental Sciences, Seoul Women's University, Seoul 01797, Korea; dkfma@swu.ac.kr (A.R.K.); bs6238@swu.ac.kr (B.S.L.); seol_jaewon@swu.ac.kr (J.S.)
2   Division of Ecological Survey Research, National Institute of Ecology, Seoul 33657, Korea; sync03@nie.re.kr
3   Department of Biology, Kongju National University, Kongju 32588, Korea; youeco21@kongju.ac.kr
4   Samseong Landscape Co., Ltd., Andong 36665, Korea; kjc3700@hanmail.net
*   Correspondence: leecs@swu.ac.kr; Tel.: +82-2-970-5666

**Abstract:** *Research Highlights*: This study was carried out to diagnose the forest ecosystem damaged by air pollution and to then develop a restoration plan to be used in the future. The restoration plan was prepared by combining the diagnostic assessment for the damaged forest ecosystem and the reference information obtained from the conservation reserve with an intact forest ecosystem. The restoration plan includes the method for the amelioration of the acidified soil and the plant species to be introduced for restoration of the damaged vegetation depending on the degree of damage. *Background and Objectives*: The forest ecosystem around the Seokpo smelter was so severely damaged that denuded lands without any vegetation appear, and landslides continue. Therefore, restoration actions are urgently required to prevent more land degradation. This study aims to prepare the restoration plan. *Materials and Methods*: The diagnostic evaluation was carried out through satellite image analysis and field surveys for vegetation damage and soil acidification. The reference information was obtained from the intact natural forest ecosystem. *Results*: Vegetation damage was severe near the pollution source and showed a reducing trend as it moved away. The more severe the vegetation damage, the more acidic the soil was, and thereby the exchangeable cation content and vegetation damage were significantly correlated. The restoration plan was prepared by proposing a soil amelioration method and the plants to be introduced. The soil amelioration method focuses on ameliorating acidified soil and supplementing insufficient nutrients. The plants to be introduced for restoring the damaged forest ecosystem were prepared by compiling the reference information, the plants tolerant to the polluted environment, and the early successional species. The restoration plan proposed the *Pinus densiflora*, *Quercus mongolica*, and *Cornus controversa–Juglans mandshurica* communities as the reference conditions for the ridge, slope, and valley, respectively, by reflecting the topographic condition. *Conclusions*: The result of a diagnostic assessment showed that ecological restoration is required urgently as vegetation damage and soil acidification are very severe. The restoration plan was prepared by compiling the results of these diagnostic assessments and reference information collected from intact natural forests. The restoration plan was prepared in the two directions of soil amelioration and vegetation restoration.

**Keywords:** air pollution; diagnostic assessment; forest ecosystem; reference information; restoration plan

## 1. Introduction

Most developed countries have decreased anthropogenic air pollution emissions by implementing abatement polices [1,2]. Korea has also practiced such a policy, and thus environmental conditions around the industrial complexes of large scale have improved greatly [3,4].

However, factories of a small scale that are far away from the public's attention, such as the Seokpo smelter where this study was carried out, still emit air pollutants and thereby cause vegetation damage and acidify soil. The forest ecosystem around the Seokpo zinc smelter was severely destroyed and therefore landslides sometimes occur. Industrial activities have resulted in the enormous emissions of air pollutants for about 40 years since the 1970s when the smelter was constructed in Seokpo in central eastern Korea. The pollutants have continued to affect the surrounding forests and other ecosystems. Forest vegetation has become sparse and poor as trees have withered, undergrowth has disappeared, and bare ground has appeared throughout the wide area.

Pollutants discharged beyond the limits of the buffering capacity of an ecosystem prevent it from maintaining its normal structure and function. Excessive land and energy use and the ecological imbalance that it brings appear to be major factors that threaten environmental stability on local as well as global levels [4–8]. The vegetation decline and subsequent soil erosion and landslides observed in the vicinity of the Seokpo zinc smelter correspond to such an example. In fact, global environmental problems such as climate change are also due to this functional imbalance between the pollution source and the sink [9–11].

If the population grows and the land and energy use continue to intensify, such ecological imbalance is likely to increase even more in the future [7,12,13]. Indeed, industrialized and urbanized areas have been expanding steadily, and the real size of degraded vegetation, such as grassland and shrubland, has increased proportionally to such land transformation in industrial areas of Korea [4,7,8,14,15]. Moreover, vegetation decline induces the structural simplification and functional weakening of plant communities, consequently leading to negative effects on ecosystem service, which provides invaluable benefits to us [12,13]. In this respect, the restoration of degraded ecosystems is urgently required to prevent the spread of such additive pollution damage [4,7,13,16].

Ecological restoration is aimed at recovering the sound natural conditions before destruction. Ecological restoration is an ecological technology that heals the damaged nature by imitating the system and function of the integrated nature, thus providing habitats for various creatures and seeking to secure the future environment of humankind [17–20]. Ecological restoration has been considered as improving ecological productivity in degraded lands, conserving biological diversity, and mitigating lost or damaged ecosystems [19–28]. Human aids are often required to restore the damaged ecosystems and prevent further damage [16,19,20,29], and provisions of extra propagules and site amendment may initiate recovery processes [30].

In order to heal the damaged nature, we must first check what problems the target has, such as how much it is degraded or what is the cause of the damage. In other words, a diagnostic assessment of the restoration target should be made [19,20].

All restoration projects are with targets to reduce the negative ecological impacts of the past and to restore the natural potential of the restoration target as much as possible. Ecological restoration means copying nature by studying a system of the integrate nature. There are several planning steps, based on the results of diagnostic assessments, to find the deficits and the targets for planning to improve the degraded ecosystem. First of all, we have to prepare such measures by obtaining diverse ecological information on an area to be restored because specific restoration efforts have to be applied in the field [31–33].

The restoration of an ecosystem damaged by environmental pollution can be achieved either through improvement of the environment polluted or by establishing plants tolerant to the pollutants [7,8,13,16,29,34–36]. Species tolerant to environmental pollution can persist through growth and reproduction or even expand their distribution range in the polluted environment [5–8,37,38].

The Seokpo smelter is uniquely located on a small mountain village and is thus far away from the public's attention. Therefore, little is known about the situation and, moreover, no academic research has been carried out. However, due to the effects of air pollutants emitted over a long period of time and the soil pollution resulting from them,

the forest damage began to become visible, and it has led to landslides and damage to surrounding rivers in good condition, causing worry in recent years. There is, therefore, a growing demand for restoration. The major pollution source of the Seokpo smelters, which refines primarily processed ore rather than raw ore, is sulfur dioxide generated from the combustion of fossil fuels used as an energy source. The damage state was similar to that of Ulsan and Yeocheon industrial complexes, major industrial complexes in Korea [4,7,8,12,36].

Restoration is an ecological technology that ameliorates degraded nature by imitating integrated and healthy nature. Restoration is achieved through a series of procedures, such as a survey of the existing conditions, a statement of the goals and objectives, the designation and description of a reference, the preparation of a master plan, the establishment of a restoration plan, restoration practices, monitoring, adaptive management, and evaluation [39–42]. Such ecological restoration is common as a means to solve such problems in developed countries, which correctly recognize that the environmental problems at a global level, such as climate change, are due to the functional imbalance between the artificial environment as an environmental stress source and the natural environment as its sink. However, in most developing countries, including Korea, most restoration projects have neglected such procedures and thus have not met the restoration goals, in spite of great expense and labor [43–50]. A series of procedures are required to achieve successful ecological restoration. However, these procedures usually tend to be ignored in most restoration projects implemented in Korea. Diagnostic evaluation is generally omitted. Even if a diagnostic evaluation is made, there are very few cases in which the level and method of restorative treatment are determined based on the results, and most restoration projects progress only by active methods without any relation to the degree of damage [46,49]. Therefore, cost and energy are wasted, and the effect is very little [46,47,50]. In most restoration projects, the reference information is not used, and restoration is performed based on the subjective decisions of the project manager. Thus, restoration projects are conducted without any model or goals. Consequently, exotic species, which should be excluded thoroughly in a restoration project, are introduced frequently, and the spatial distribution range for plant species is barely considered [46,47,50]. Therefore, most restoration projects remain at the level of past afforestation or classical landscaping.

This study was attempted to implement ecological restoration of the advanced level, which is beyond this low level of restoration. This study conducted a diagnostic evaluation for the damaged forest from air and soil pollution, collected reference information from intact forest without any damage, and prepared a restoration plan by combining such information. In order to ecologically restore the area, the following situations need to be considered: first, since pollutants continue to be emitted, tolerant plants that can withstand such pollution should be selected and introduced. Second, soils contaminated by the effects of pollutants discharged for a long time should be improved. Third, many species have already disappeared due to environmental pollution, and thus we need to introduce species that have disappeared based on the ecological information obtained from the reference ecosystem. Finally, since the environment has been severely damaged to the point where landslides occur, measures should also be taken to prevent landslides when establishing the planting bed. This study aims to clarify the extent and the type of damaged forest and, furthermore, to recommend a restoration plan suitable for the ecological condition of the target site as well as the damage degree based on the principle of restoration ecology. In order to arrive at these goals, we assessed vegetation damage based on satellite image interpretation and field checks. Vegetation damage was also assessed based on species composition and species diversity. Furthermore, we also diagnosed the damaged state of soil based on its physic-chemical properties. We recommended the restoration plan by synthesizing the results of the diagnostic assessment and the reference information, including pollution tolerant species and early successional species considering the environmental condition of this area.

## 2. Materials and Methods

### 2.1. Study Site

This study was carried out in the forest ecosystem around the Seokpo smelter, located on the central eastern Korea (Figure 1). The Seokpo smelter has been in operation since 1970. The smelter produces zinc, cadmium sulfate, copper sulfate, and manganese sulfate and has emitted many air pollutants around it [51]. This area is surrounded by mountainous areas with steep slopes and has topographical features that make it difficult for air pollutants to spread (Figure 2) [4,6,16]. In addition, the temperature inversion, which occurs as the cold air on the mountain descends along the surrounding mountain slopes and is trapped at the bottom of the basin after sunset, makes it more difficult to spread air pollutants, causing an increase in pollution damage in this narrow valley. During this temperature inversion time, dense smoke often settles in low-lying areas and becomes trapped due to temperature inversions—when a layer within the lower atmosphere acts as a lid and prevents vertical mixing of the air. Steep canyon walls act as a horizontal barrier, concentrating the smoke within the deepest parts of the canyon and increasing the strength of the inversion [52,53].

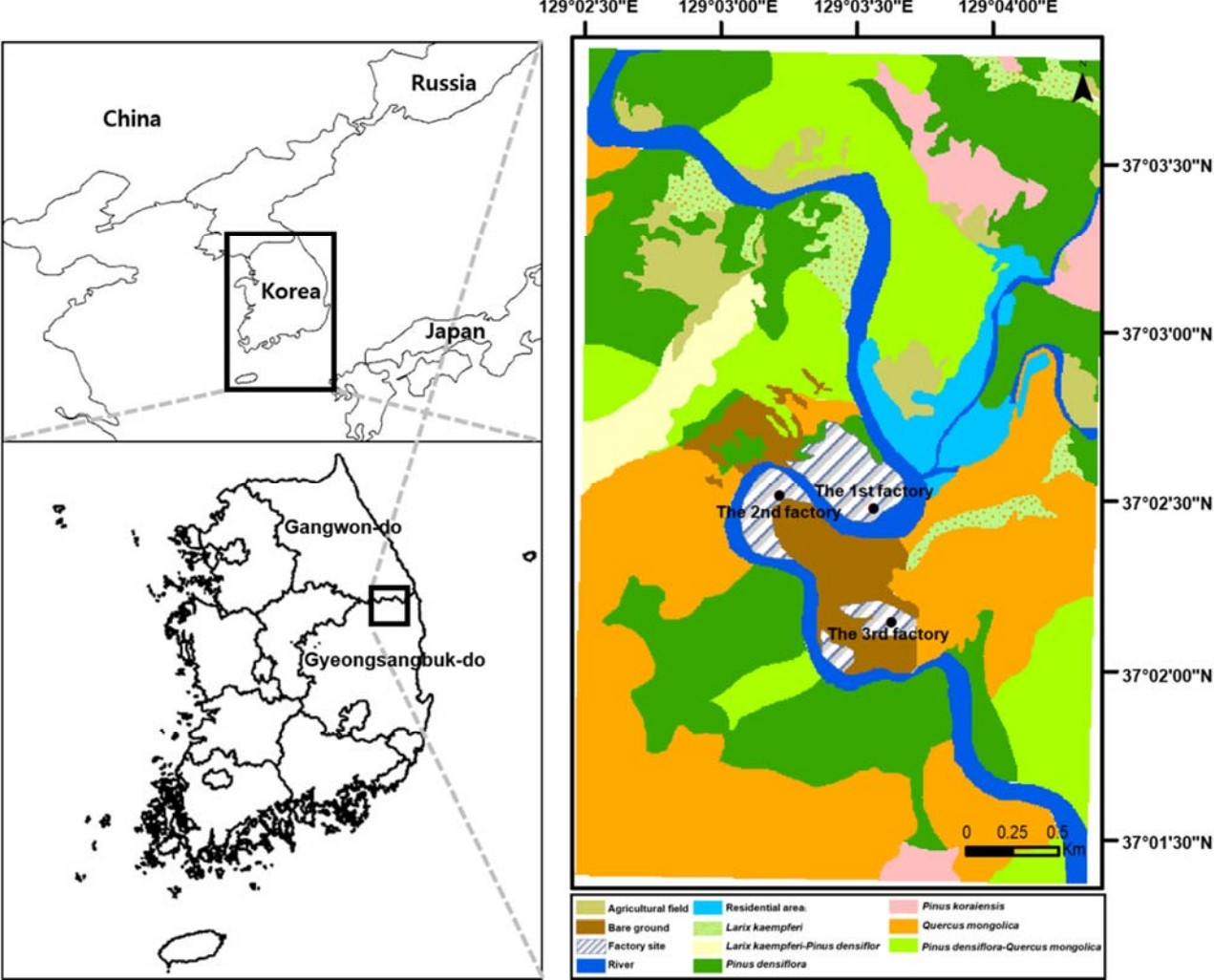

**Figure 1.** A map showing the study area, the Seokpo zinc smelter, which is located in central eastern Korea. Seokpo zinc smelter is composed of three factories. A colored map shows vegetation and land use types established around the smelter. Dots and the parts expressed with oblique lines around them indicate factories and factory sites.

As is shown in a vegetation map in Figure 1, the vegetation of this area is dominated by the *Quercus mongolica* community. However, the *Pinus densiflora* community is established

on the slopes of mountainous areas with steep slopes or mountain ridges and peaks with shallow soil depth due to edaphic characteristics. There is also a mixed forest in which two species forming a community together. On the other hand, there are plantations where *Larix kaempferi* and *P. koraiensis* are introduced artificially in some areas, and there are places where mixed forests are formed where natural vegetation is mixed with planted species. Meanwhile, agricultural and residential areas are established in the lowlands.

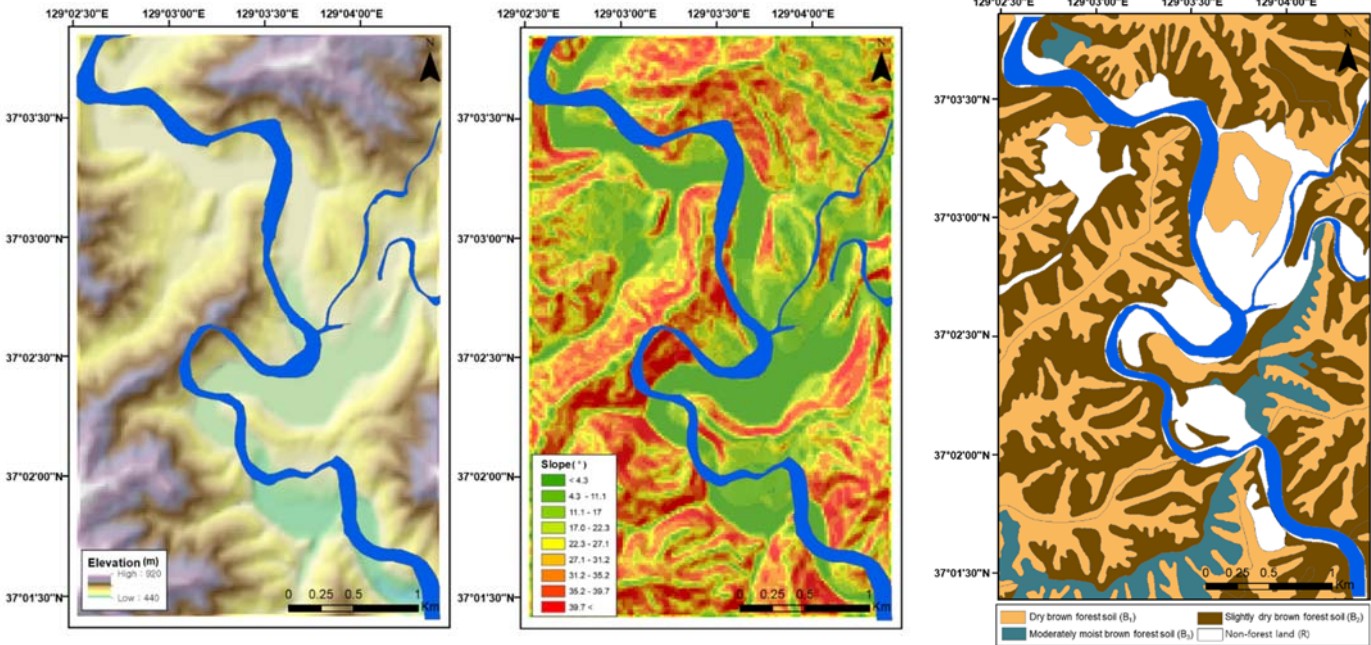

**Figure 2.** Maps showing the spatial distribution of elevation (m), slope (degree), and soil type in the study area.

The climate of Bonghwa is continental, with warm and moist summers and cold and dry winters. The mean annual temperature is 9.9 °C and the high and low mean temperatures are recorded as 28.6 °C and −10.3 °C in August and January, respectively. The mean annual precipitation is 1217.9 mm; about 60% falls in the rainy season from June to August and, including the typhoon season of September, about 70% is concentrated in both periods [54].

The elevation of the study area ranges from 400 to 900 m above sea level. The slope degree is as steep as more than 20° in most of the mountainous land except the valley. The parent rock of the study area consists mostly of granite, and in the flat land beside rivers and streams consists of alluvium. Soil in this area is composed of dry (B1), slightly dry (B2), and moderately moist brown forest soil (B3), which were developed on granite bedrock [55] (Figure 1).

The reference forest, used for comparison, was designated as the Korean red pine (*Pinus densiflora* Siebold & Zucc.) and oak (*Quercus mongolica* Fisch. Ex Ledeb., and *Q. variabilis* Blum communities), which are the representative late successional vegetation types in Korea, and *Cornus controversa* Hemsl. ex Prain, which represents the valley forest [56]. The reference forests were selected in the Uljin genetic resource reserve, which are about 15 km from this study area and therefore retain a healthy vegetation. The reference forests were selected as the forests that are from 50 to 100 years old, which is not an old growth forest but a stable forest. The number of plots chosen for the survey for the reference forests was 10, 10, and 10 for the *P. densiflora*, *Q. mongolica*, and *Cornus controversa* communities, respectively.

*2.2. Methods*

A vegetation map was made based on image interpretation and field checks. Aerial photo images (1:5000 scale) were used to identify the vegetation types and boundaries, which appear as a homogeneous patch. These vegetation types were confirmed by field checks. Vegetation types were overlapped onto topographical maps at a 1:5000 scale. Patches smaller than 1 mm on the map were excluded from this study because of the uncertainty of their sizes and shapes [57]. The final map was constructed with ArcGIS program (ver. 10.0, ESRI, Redlands, CA, USA) [58].

To determine the vitality of vegetation in the study area, Landsat images taken on 2 October 2018 were downloaded to analyze the normalized differential vegetation index (NDVI). Vegetation damage based on vitality was assessed through supervising analysis on the satellite image [59]. This study applied a supervised classification-maximum likelihood algorithm to classify the vegetation damage state around the Seokpo zinc smelter using Landsat images in the ArcGis10.1 program. The maximum likelihood algorithm is the most common method in remote sensing image data analysis [60], which is mainly controlled by selecting the pixels that are representative of the desired classes [61]. Using the signature file creation tool, vegetation damage was classified into five classes of very severe, severe, moderate, light, and none. The damage degrees classified were verified through field checks as follows.

Visible damage was investigated by recording the degree of necrosis that appeared on the leaf surface of plants appearing in the process of the vegetation survey. The damage degree was classified into five groups based on the percentage of injury shown on the leaf surface: very severe (V, more than 75% of total leaf area damaged), severe (IV 50–75% damaged), moderate (III, 25–50% damaged), light (II, less than 25% damaged), and none (I, 0%) [7].

The vegetation structure damage was assessed by the deformation of vegetation stratification based on [62]. More than 50% of the land, which is covered with barren ground was assessed as 'very severe'. Grassland or barren ground without any woody plants was assessed as 'severe'. Vegetation that had lost some stratification was assessed as 'moderate'. Vegetation with visible damage only to integrate structure, with all strata composed of canopy, understory, shrub, and herb layers shown without any loss of stratification was assessed as 'light'. A map expressing vegetation damage was prepared by applying the GIS program (ver. 10.0, ESRI, Redlands, CA, USA).

The vegetation survey was carried out from May to September in 2018 and 2019. The vegetation survey was carried out by recording the Braun–Blanquet's cover class of plant species appearing in quadrats of 2 m × 2 m, 5 m × 5 m, and 20 m × 20 m size in grassland, shrubland, and forest, respectively, installed randomly [63]. The vegetation survey was carried out in 99 plots (28, 28, 1, 5, 5, 19, and 13 plots for *Pinus densiflora* community, *Quercus mongolica* community, *Q. variabilis* community, valley forest, shrubland, grassland, and cut slope, respectively) from May to September in 2018 and 2019.

Soil samples were collected with a sampling spade in June–August 2019 from the top 10 cm after removing the litter at five random points in each plot, after which they were pooled, air dried at room temperature, and sieved through 2 mm mesh. A total of 18, 18, 1, 3, 3, 12, and 9 soil samples were collected from the *Pinus densiflora* community, the *Quercus mongolica* community, the *Q. variabilis* community, valley forest, shrubland, grassland, and cut slope, respectively. Soil properties were diagnosed for pH and $Ca^{2+}$, $Mg^{2+}$, and $Al^{3+}$ content. Soil pH was measured with a bench top probe after mixing the soil with distilled water (1:5 ratio, w/v) and filtering the extract (Whatman No. 44 paper). Organic matter (OM) concentration was estimated by loss of dry mass on ignition at 400 °C. Total nitrogen was measured with the micro-Kjeldahl method [64]. Available P was extracted in 1-N ammonium fluoride (pH = 7.0) and exchangeable $Ca^{2+}$, $Mg^{2+}$, and $Al^{3+}$ contents were extracted with 1N ammonium acetate (pH = 7.0 for Ca and Mg and pH = 4.0 for Al) and measured by ICP (inductively coupled plasma atomic emission spectrometry; Shimadzu ICPQ-1000) [65]. The results of the analysis on the physic-chemical properties of soil were

reinforced by the simple kriging model. Maps expressing the physic-chemical properties of the soil were prepared by applying the GIS program (Version 10.1). The soil properties (pH, OM, N, P, Ca, Mg and Al) of sites showing different degrees of damage to vegetation were compared with one-way analysis of variance (ANOVA) and Tukey's honestly significant difference (HSD) test at $\alpha = 0.05$ [66].

The restoration plan was prepared by recommending a soil amelioration method and the plant species to be introduced depending on the degree of damage to the vegetation and soil. Dolomite and organic fertilizer were recommended for soil amelioration. The dolomite requirement was calculated by applying the following equation: dolomite requirement (t/ha) = (target pH–current pH) × soil texture factor. We decided to set the target pH as 5.5, based on the normal pH of the natural forest soil, and soil texture factor as 3, reflecting the soil texture of this area [67]. Dolomite raises the soil pH and increases available $Ca^{2+}$ and $Mg^{2+}$ due to the following chemical reactions in the soil solution [68]:

$$\text{Initial chemical reaction: } Ca \bullet Mg(CO^3)_2 + 2H^+ \rightarrow 2HCO_3{}^- + Ca^{2+} + Mg^{2+}, \tag{1}$$

$$\text{Second reaction: } 2HCO_3{}^- + 2H^+ \rightarrow 2CO^2 + 2H_2O, \tag{2}$$

$$\text{Net reaction: } Ca \bullet Mg(CO^3)_2 + 4H^+ \rightarrow Ca^{2+} + Mg^{2+} + 2CO^2 + 2H_2O, \tag{3}$$

The amount of organic fertilizer applied was determined to be half the level of dolomite, referring to previous study results [8]. The chemical characteristics of the organic fertilizer are given in Appendix A, Table A1.

The introduction of vegetation for restoration took the form reinforcing the lost part in the vegetation stratification. Therefore, we planned to introduce the disappeared species compared to the species composition of the natural reference site. Furthermore, we added tolerant and early successional species in our restoration plan, considering the environmental condition of this area, where air and soil pollution continues and vegetation is so severely damaged that bare ground can appear and severe soil erosion occurs. The plant species to be introduced were selected by applying indicator species analysis. Indicator species analysis was carried out using the function 'vegan', 'indicspecies' of the R statistical package (version 4.0.2). In addition, we reinforced the plant species to be introduced by referring to national vegetation information [69] and existing research data conducted on the reference site of this study [70], considering that this study was conducted in a limited place.

## 3. Results

### 3.1. Vegetation Damage

The spatial distribution of NDVI showed that its value was lower near the factory and tended to increase as it moved away from it (Figure 3). The value was also related to the topographic condition; thus, it was low in the ridge and high in the valley (Figure 3).

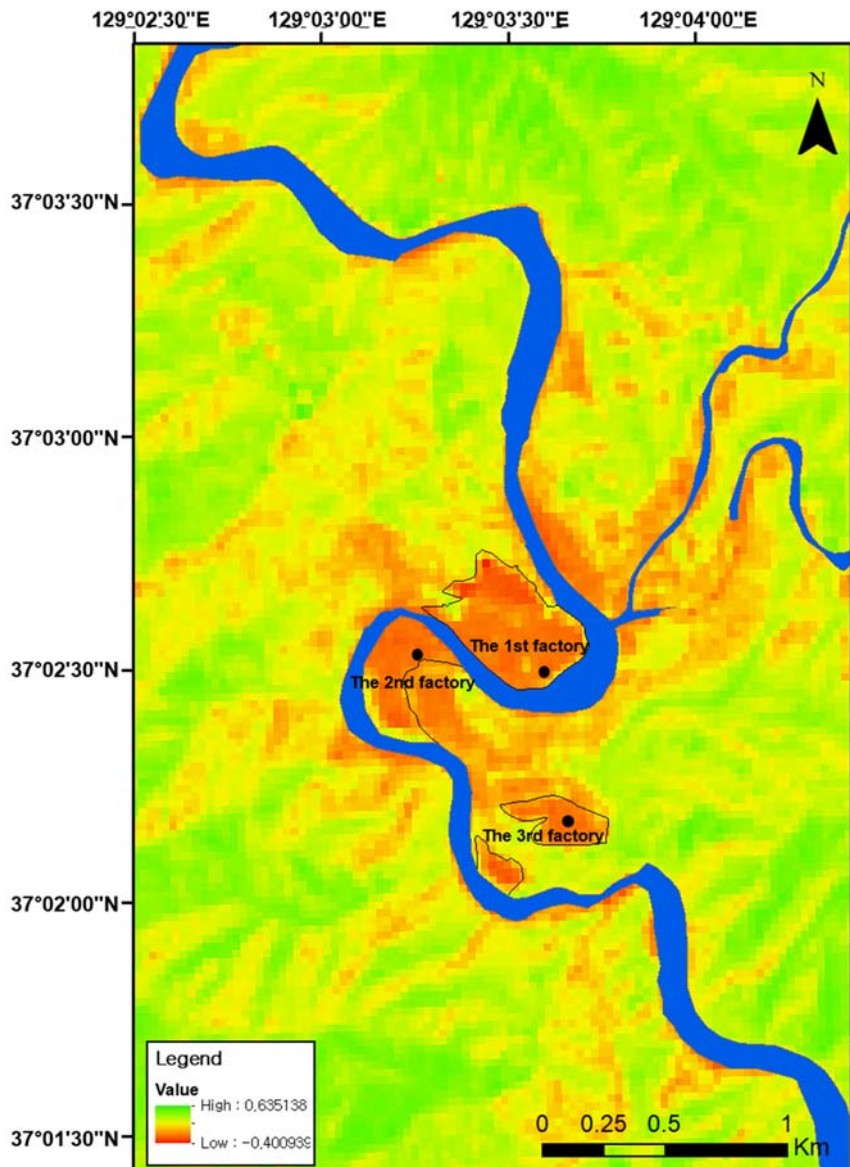

**Figure 3.** Spatial distribution of NDVI in the study area.

Vegetation damage identified from the satellite image interpretation depended on the distance from the smelter and topography. The damage appeared was severer in sites closer to the smelter and decreased farther away (Figure 4). The degree of damage was also dominated by topographic conditions, and therefore damage was restricted within the first ridge from the pollution source, little damage thus appearing on the opposite slope or beyond the first ridge from the smelter (Figure 4).

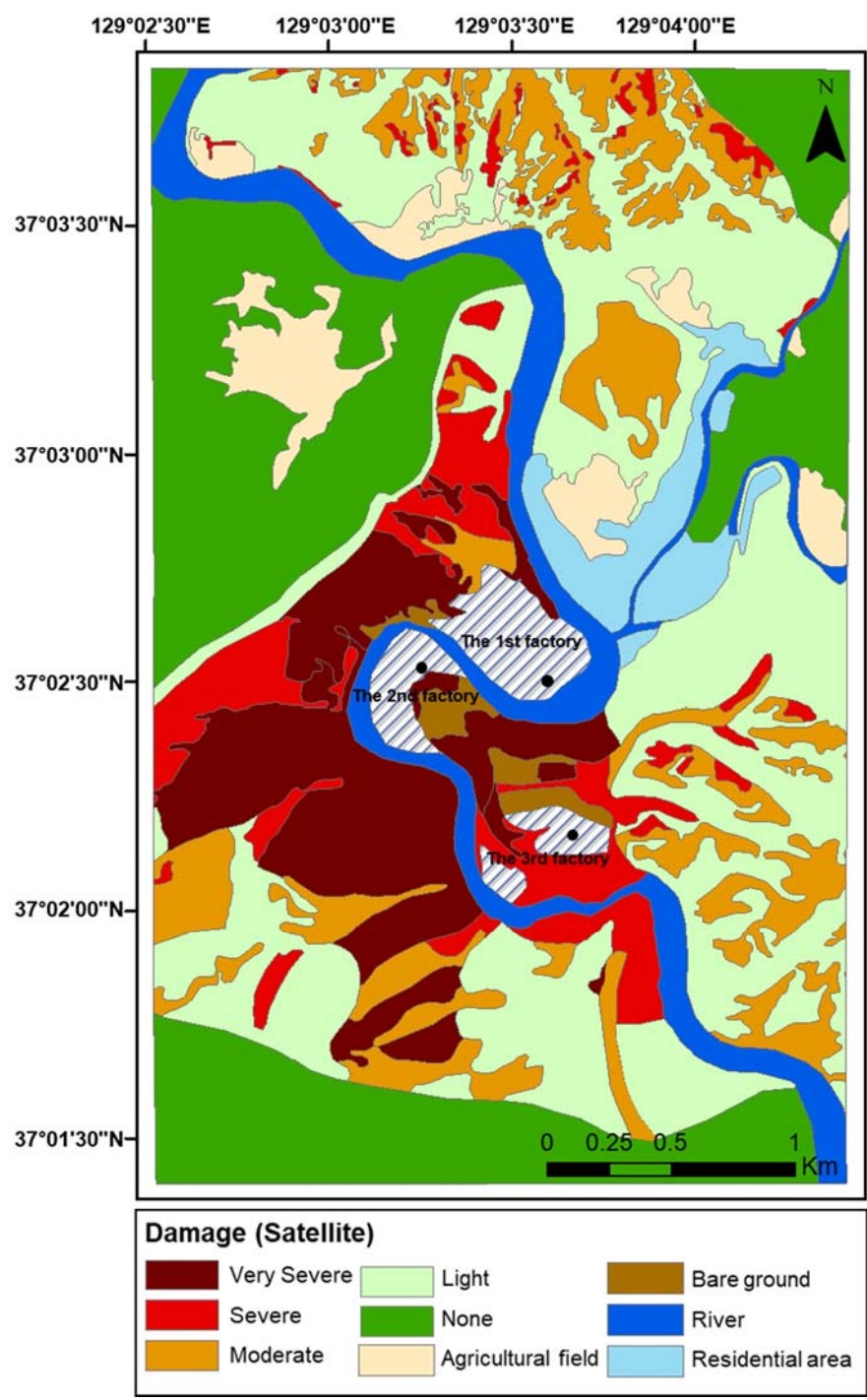

**Figure 4.** Spatial distribution of vegetation damage based on satellite image interpretation in the study area.

Damage based on vegetation stratification showed a trend similar to the abovementioned results. Vegetation in the site where the damage was light showed the integrate structure with all strata composed of canopy, understory, shrub, and herb layers. However, vegetation structure became simplified with the increase of the damage, and thus grassland or barren ground appeared in sites where the damage was the severest (Figure 5).

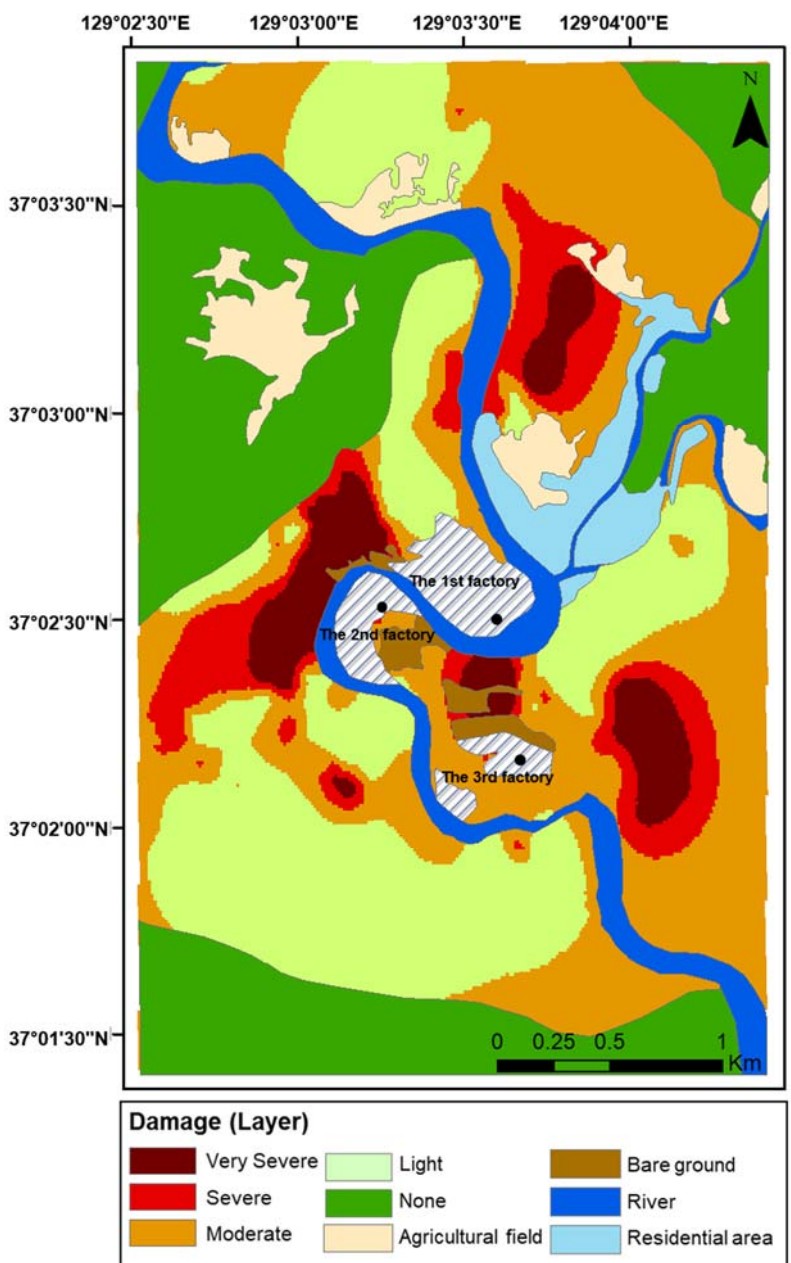

**Figure 5.** Spatial distribution of vegetation damage based on vegetation stratification in the study area.

In a map where vegetation damage by damage class is shown (Figure 4), very severely damaged vegetation appeared in areas located in the northwestern direction of the first factory, the western and southern directions of the second and third factory, and surrounded by the first, second, and third factories. Severely damaged vegetation appeared in the areas farther than the very severely damaged vegetation from the three factories in all four directions of east, west, south, and north. Moderately damaged vegetation appeared in the areas located in the eastern and western directions, with the third factory at the center. Lightly damaged vegetation appeared in the areas far from them within the first ridge from the factories.

### 3.2. Soil Degradation

Spatial distribution of the physic-chemical properties of the soil reflected a trend of vegetation damage. Soil pH was usually low compared with the unpolluted area but was lower in sites close to pollution sources and became higher farther away (Figure 6).The

$Ca^{2+}$ and $Mg^{2+}$ content showed trends similar to that of the pH, whereas $Al^{3+}$ content represented a reverse trend (Figure 6).

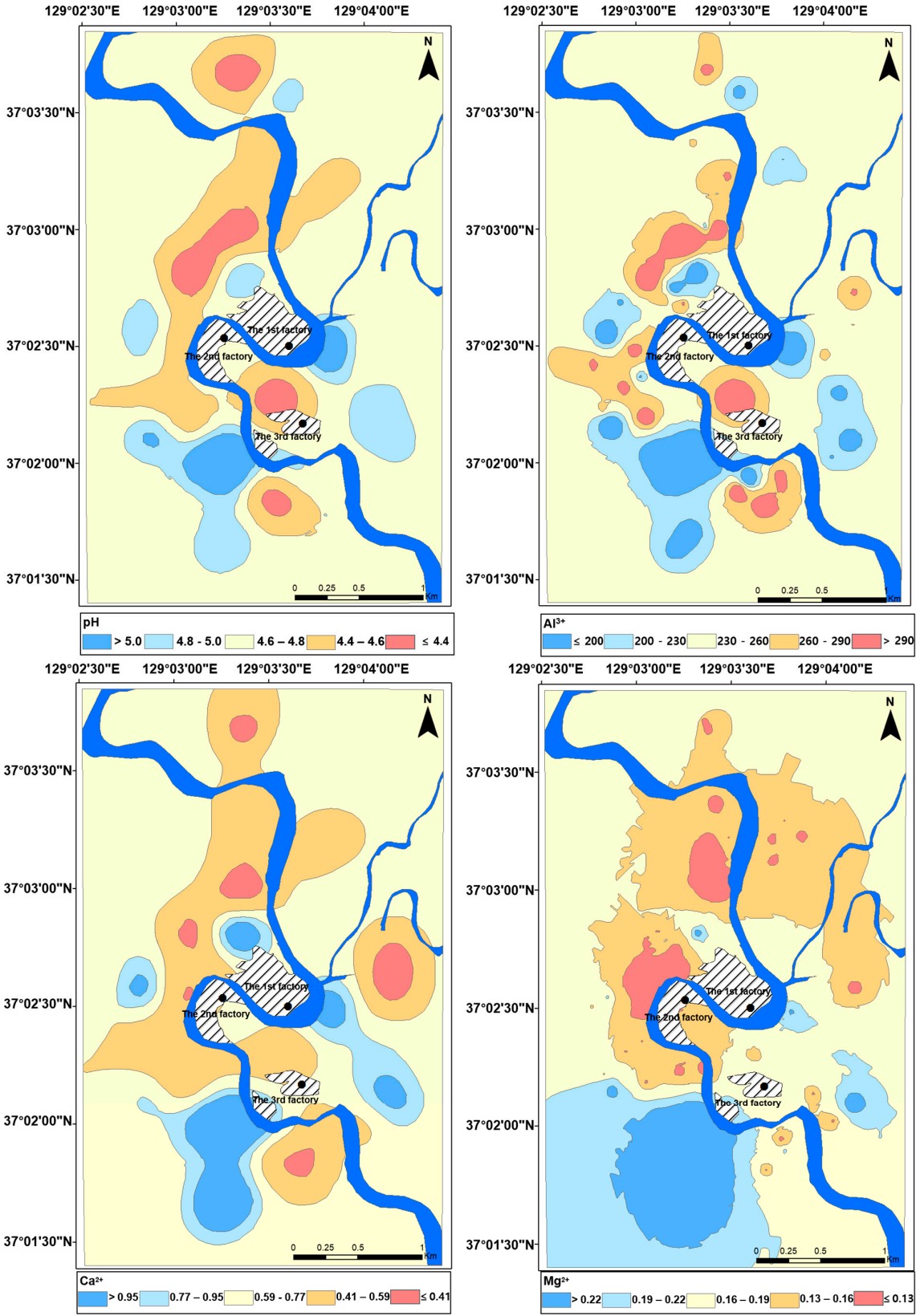

**Figure 6.** Spatial distribution of soil pH and $Ca^{2+}$, $Mg^{2+}$, and $Al^{3+}$ contents of soil in the study area.

Soil pH tended to be relatively low in the northern and western directions of the three factories, the area surrounded by those factories, and the southern direction of the third factory, whereas it was relatively high in the southwestern and northeastern directions of the third factory (Figure 6).

The $Ca^{2+}$ and $Mg^{2+}$ content showed trends similar to that of the pH, while $Al^{3+}$ content represented a reverse trend (Figure 6).

The physic-chemical properties of the soil were compared with those of the reference site and among the degrees of damage to the vegetation (Figure 7). The pH and $Ca^{2+}$, $Mg^{2+}$, and available phosphorus contents were lower than those in the reference site, whereas the total nitrogen content was vice versa. However, organic matter and $Al^{3+}$ content did not show any significant difference between both sites. On the other hand, pH and $Ca^{2+}$ and $Mg^{2+}$ contents showed significant difference among the degrees of damage to the vegetation, but the other factors did not show any significant differences among the degrees of damage.

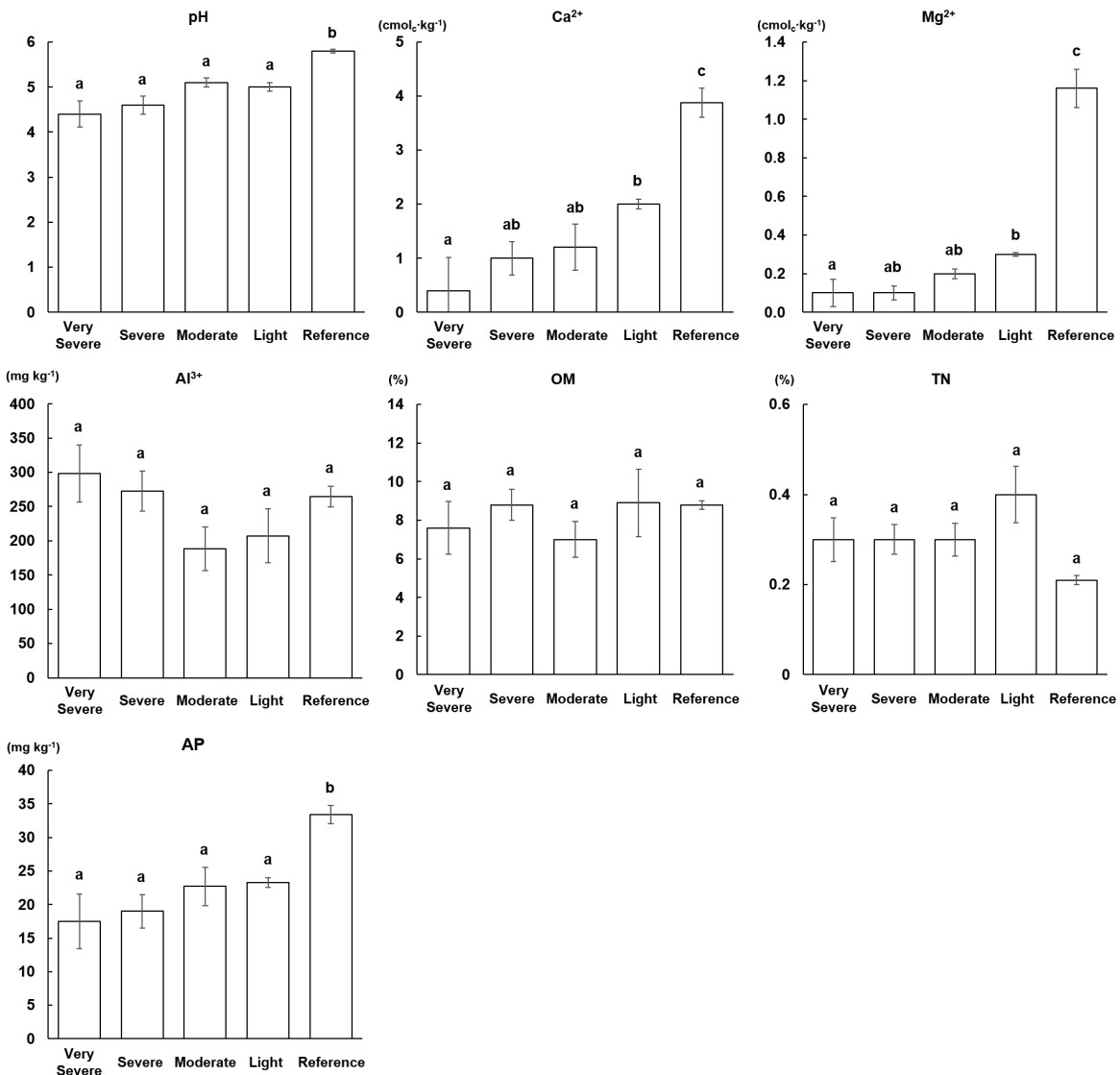

**Figure 7.** A comparison of the physic-chemical properties of the soil among damage degrees of vegetation and with that of the reference site. Very severe, severe, moderate, and light indicate damage degree and reference indicates the unpolluted site selected for comparison. OM: organic matter; TN: Total nitrogen; AP: available phosphorus. Each bar was expressed with mean and standard error of mean. Tukey's honestly significant difference (HSD) test was conducted on each of the parameters that show a statistically significant difference among the four types of damage degrees at $\alpha = 0.05$; the means with the same alphabetical character (in superscript), for each parameter, were not different from each other.

### 3.3. Species Composition

As the result of stand ordination, arrangement of stands reflected vegetation damage (Figure 8). The reference stands were arranged on the left on the AXIS I and very severely or severally damaged stands on the right, and moderately and lightly damaged stands were arranged between both groups. Moderately and lightly damaged stands tended to be arranged depending on the topographical position as *P. densiflora* stands, *Q. Mongolica* stands, and stands established in the valley were arranged in the mentioned order as moves from bottom upward on the AXIS II. Meanwhile, cut slope stands were arranged in the left part on the AXIS I like the reference stands but separated from them on the AXIS II.

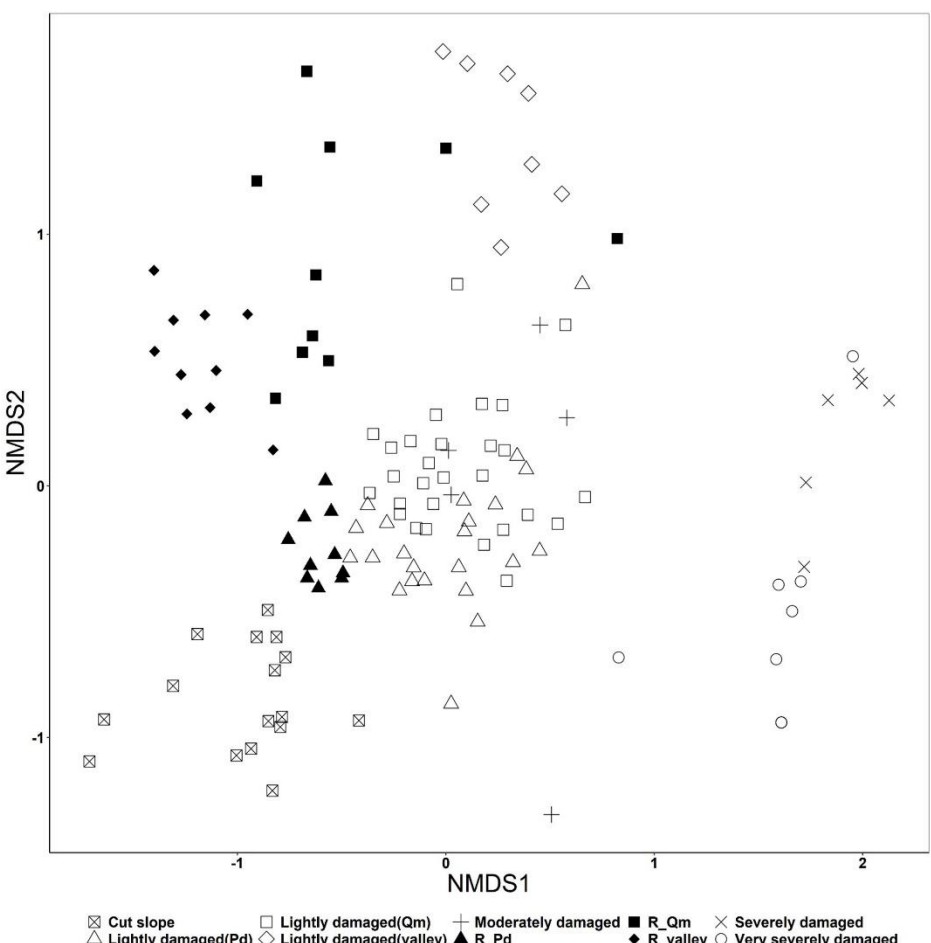

**Figure 8.** Ordination of vegetation stands established around the Seokpo smelter and on the natural reference forest, Uljin Forest Genetic Resources Conservation Reserve, central eastern Korea. Legends expressed as damage degree represent the damaged stands established around the Seokpo smelter. R_Pd: *Pinus densiflora* stands established in the reference site; R_Qm: *Quercus mongolica* stands established in the reference site; R_Valley: stands established in the reference site; Cut slope: stands established on the incised slope along the forest road ($p = 0.001$, stress = 0.1702836).

### 3.4. Species Diversity

As a result of comparing the species diversity by the species rank–dominance curve, the species diversity of the damaged sites was lower than that in the reference sites (Figure 9). In the damaged sites, species diversity tended to decrease in proportion to the damage degree (Figure 8). The species diversity was also dominated by the topographic condition and thus high in the vegetation established in the valley and low in the pine forest on the ridge, and the species diversity of broad-leaved forests on the mountain slope was located between both vegetation types (Figure 9).

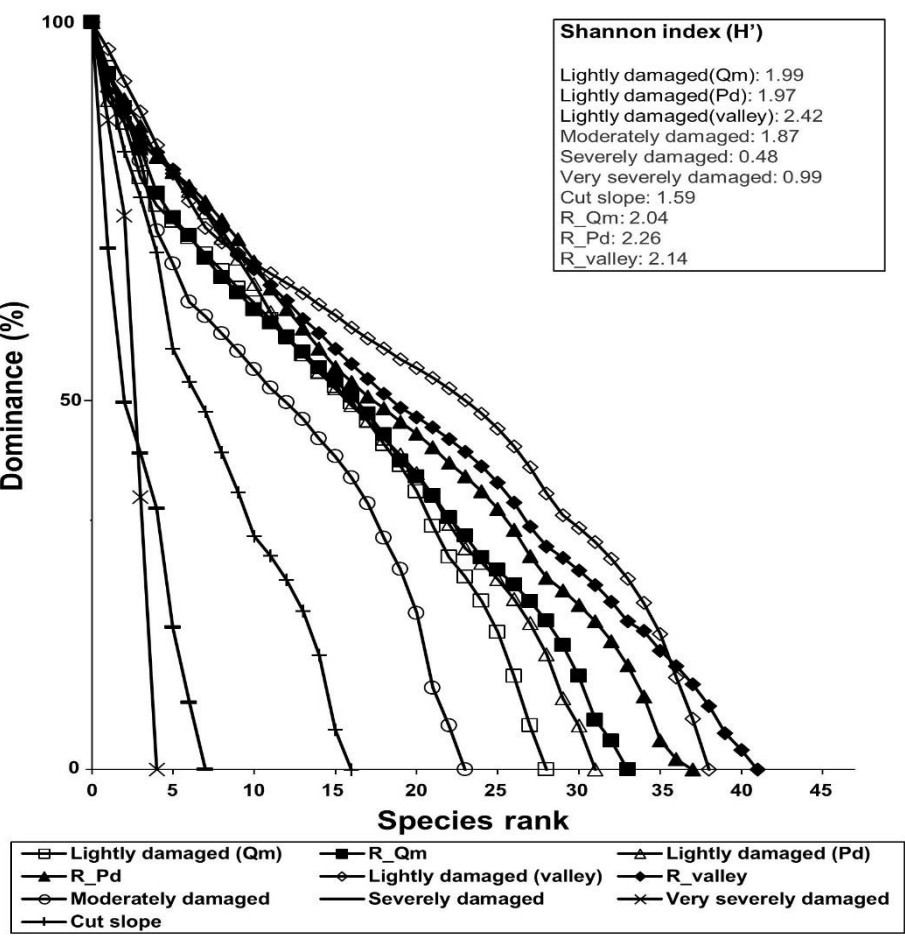

**Figure 9.** Species rank–dominance curves of vegetation stands established around the Seokpo smelter and on the natural reference forest, Uljin Forest Genetic Resources Conservation Reserve, southeastern Korea. Legends are the same as those in Figure 8.

*3.5. Selection of Plant Species for Vegetation Restoration*

We selected disappeared, tolerant, and early successional species by applying the indicator species analysis (Tables 1–4). First, we selected species to be introduced for vegetation restoration by comparing all vegetation data between polluted and natural reference sites and cut slope. We selected species which appear in the natural reference site but do not appear in the polluted site as the disappeared species. The disappeared species mean species that should be introduced for vegetation restoration. We selected species which showed the reverse trend to the disappeared species as the tolerant species. The species which appear characteristically on the cut slope were selected as the early successional species. Ecological restoration should copy the environment by studying a system of the integrate nature. However, considering the environmental condition of this area where air and soil pollution continues and vegetation is so severely damaged that bare ground can appear and severe soil erosion occurs, we added the tolerant and early successional species in our restoration plan. In addition, we added the plant species to be introduced by referring to the existing research data conducted around this study area [69,70] to enhance the stability of the restoration plan.



**Table 1.** The result of indicator species analysis for selecting the disappeared species, tolerant species to the polluted environment, and early successional species based on data collected in all study sites. Cut slope: early successional species; Severe, Moderate, Light: tolerant species in severely, moderately, and lightly damaged sites, respectively; Valley: tolerant species in valley; Reference(Pd), Reference(Qm), Reference(VA): disappeared species in the natural reference sites of *Pinus densiflora* forest, *Quercus mongolica* forest, and valley forest, respectively.

| Species Name | Site Type | Stat | *p*-Value |
|---|---|---|---|
| *Betula schmidtii* | Cut slope | 0.675 | 0.001 *** |
| *Calamagrostis arundinacea* | Cut slope | 0.787 | 0.001 *** |
| *Lindera obtusiloba* | Moderate | 0.802 | 0.001*** |
| *Tripterygium regelii* | Moderate | 0.436 | 0.009 ** |
| *Rhododendron schlippenbachii* | Light | 0.463 | 0.006 ** |
| *Actinidia arguta* | Valley | 0.525 | 0.002 ** |
| *Aralia elata* | Valley | 0.482 | 0.012 * |
| *Deutzia parviflora* | Valley | 0.502 | 0.002 ** |
| *Fraxinus rhynchophylla* | Valley | 0.414 | 0.026 * |
| *Schisandra chinensis* | Valley | 0.396 | 0.037 * |
| *Athyrium yokoscense* | Very severe | 0.904 | 0.001 *** |
| *Miscanthus sinensis* var. *purpurascens* | Severe | 0.952 | 0.001 *** |
| *Betula chinensis* | Reference(Pd) | 0.496 | 0.004 ** |
| *Disporum smilacinum* | Reference(Pd) | 0.585 | 0.001 *** |
| *Fraxinus sieboldiana* | Reference(Pd) | 0.677 | 0.001 *** |
| *Aster scaber* | Reference(Qm) | 0.513 | 0.007 ** |
| *Athyrium vidalii* | Reference(Qm) | 0.449 | 0.005 ** |
| *Callicarpa japonica* | Reference(Qm) | 0.470 | 0.005 ** |
| *Carex siderosticta* | Reference(Qm) | 0.440 | 0.007 ** |
| *Hydrangea serrata* f. *acuminata* | Reference(Qm) | 0.425 | 0.016 * |
| *Potentilla freyniana* | Reference(Qm) | 0.564 | 0.002 ** |
| *Styrax obassis* | Reference(Qm) | 0.417 | 0.030 * |
| *Carex humilis* var. *nana* | Reference(VA) | 0.441 | 0.006 ** |
| *Cornus controversa* | Reference(VA) | 0.874 | 0.001 *** |
| *Juglans mandshurica* | Reference(VA) | 0.402 | 0.050 * |

*** significant at 0.1% level, ** significant at 1% level, * significant at 5% level.

**Table 2.** The result of indicator species analysis for selecting the disappeared species and tolerant species to the polluted environment based on data collected in both polluted and natural *Pinus densiflora* forests. Polluted: tolerant species in the polluted site of *Pinus densiflora* forest; Reference: disappeared species in the natural reference site of *Pinus densiflora* forest.

| Species Name | Site Type | Stat | *p*-Value |
|---|---|---|---|
| *Athyrium yokoscense* | Polluted | 0.315 | 0.039 * |
| *Lindera obtusiloba* | Polluted | 0.447 | 0.004 ** |
| *Miscanthus sinensis* var. *purpurascens* | Polluted | 0.333 | 0.040 * |
| *Quercus mongolica* | Polluted | 0.446 | 0.011 * |
| *Rhododendron schlippenbachii* | Polluted | 0.461 | 0.024 * |
| *Atractylodes ovata* | Reference | 0.418 | 0.033 * |
| *Betula chinensis* | Reference | 0.391 | 0.034 * |
| *Carex humilis* var. *nana* | Reference | 0.704 | 0.001 *** |
| *Dendranthema zawadskii* var. *latilobum* | Reference | 0.419 | 0.031 * |
| *Disporum smilacinum* | Reference | 0.568 | 0.001 *** |
| *Fraxinus sieboldiana* | Reference | 0.533 | 0.002 *** |
| *Lespedeza bicolor* | Reference | 0.886 | 0.001 *** |
| *Rhododendron micranthum* | Reference | 0.634 | 0.001 *** |
| *Rhododendron mucronulatum* | Reference | 0.422 | 0.031 * |

*** significant at 0.1% level, ** significant at 1% level, * significant at 5% level.

**Table 3.** The result of indicator species analysis for selecting the disappeared species and tolerant species to the polluted environment based on data collected in both polluted and natural *Quercus mongolica* forests. Polluted: tolerant species in the polluted site of *Quercus mongolica* forest; Reference: disappeared species in the natural reference site of *Quercus mongolica* forest.

| Species Name | Site Type | Stat | *p*-Value |
|---|---|---|---|
| *Athyrium yokoscense* | Polluted | 0.310 | 0.034 * |
| *Fraxinus sieboldiana* | Polluted | 0.419 | 0.008 ** |
| *Lindera obtusiloba* | Polluted | 0.514 | 0.001 *** |
| *Rhododendron schlippenbachii* | Polluted | 0.498 | 0.004 ** |
| *Acer pseudosieboldianum* | Reference | 0.397 | 0.006 ** |
| *Ainsliaea acerifolia* | Reference | 0.543 | 0.001 *** |
| *Artemisia keiskeana* | Reference | 0.449 | 0.002 ** |
| *Aster scaber* | Reference | 0.407 | 0.018 * |
| *Athyrium vidalii* | Reference | 0.351 | 0.015 * |
| *Atractylodes ovata* | Reference | 0.448 | 0.001 *** |
| *Betula schmidtii* | Reference | 0.380 | 0.012 * |
| *Callicarpa japonica* | Reference | 0.419 | 0.015 * |
| *Carex humilis* var. *nana* | Reference | 0.419 | 0.016 * |
| *Carex siderosticta* | Reference | 0.341 | 0.002 ** |
| *Hydrangea serrata* f. *acuminata* | Reference | 0.360 | 0.015 * |
| *Lespedeza bicolor* | Reference | 0.306 | 0.027 * |
| *Polystichum tripteron* | Reference | 0.391 | 0.017 * |
| *Potentilla freyniana* | Reference | 0.645 | 0.001 *** |
| *Rubus crataegifolius* | Reference | 0.419 | 0.015 * |

*** significant at 0.1% level, ** significant at 1% level, * significant at 5% level.

**Table 4.** The result of indicator species analysis for selecting the disappeared species and tolerant species to the polluted environment based on data collected in both polluted and natural valley forests. Pol-luted: tolerant species in the polluted site of valley forest; Reference: disappeared species in the natural reference site of valley forest.

| Species Name | Site Type | Stat | *p*-Value |
|---|---|---|---|
| *Actinidia arguta* | Polluted | 0.423 | 0.028 * |
| *Athyrium yokoscense* | Polluted | 0.548 | 0.019 * |
| *Lindera obtusiloba* | Polluted | 0.655 | 0.001 *** |
| *Miscanthus sinensis* var. *purpurascens* | Polluted | 0.554 | 0.028 * |
| *Quercus mongolica* | Polluted | 0.432 | 0.038 * |
| *Cornus controversa* | Reference | 0.759 | 0.001 *** |
| *Rhododendron mucronulatum* | Reference | 0.496 | 0.048 * |

*** significant at 0.1% level, * significant at 5% level.

### 3.6. Zonning and Design for Restorative Treatment

The spatial range for which the restoration was required was restricted within the first ridge from the pollution source, considering the impact extent of the air pollution. We divided the district for restoration into four zones of very severely, severely, moderately, and lightly damaged zones based on the damage degree (Table 5) and three zones of ridge, slope, and valley based on the topographic conditions (Table 6) [13].

**Table 5.** Level and method of restoration recommended based on a diagnostic evaluation of the forest ecosystem around Seokpo zinc smelter, central eastern Korea.

| Damaged Degree | Vegetation Status | Soil pH | Restoration Method |
|---|---|---|---|
| Very severe | Grassland with low coverage or bare ground | 4.4 | Soil amelioration: dolomite 4.5 ton/ha + organic fertilizer 2.3 ton/ha<br>Introduction of plants forming all layers of vegetation |
| Severe | Canopy layer disappeared and shrub and herb layers are poor | 4.6 | Soil amelioration: dolomite 3.0 ton/ha + organic fertilizer 1.5 ton/ha<br>Introduction of plants forming canopy tree, shrub, and herb layers |
| Moderate | All vegetation strata exist but coverage is poor | 5.1 | Soil amelioration: dolomite 1.5 ton/ha + organic fertilizer 0.8 ton/ha<br>Introduction of plants forming shrub and herb layers |
| Light | Development of undergrowth is poor | 5.0 | Soil amelioration: dolomite 1.0 ton/ha + organic fertilizer 0.5 ton/ha<br>Vegetation restoration is left to passive restoration |

**Table 6.** Species to be introduced for restoration by layer of vegetation in each topographic condition. This species information was prepared by incorporating disappeared species compared with the natural reference site, tolerant species to the polluted environment, and early successional species. In very severely and severely damaged zones, plant species forming all layers of vegetation stratification including canopy tree, understory tree, shrub, and herb layers are recommended for restoration. Plant species forming shrub and herb layers are recommended in moderately damaged zone. Passive restoration is recommended in lightly damaged zone. Vegetation type, such as Korean red pine forest established as an edaphic climax type on the upper slope, usually lacks understory tree layer due to topographic condition, which is dry and infertile.

| Vegetation Stratum | Ridge | Slope | Valley |
|---|---|---|---|
| **Canopy tree layer** | *Betula schmidtii*<br>*Betula chinensis* *<br>*Pinus densiflora*<br>*Quercus variabilis*<br>etc. | *Betula davurica* *<br>*Betula schmidtii* *<br>*Quercus aliena* *<br>*Quercus mongolica*<br>*Quercus variabilis*<br>etc. | *Acer pictum* subsp. *Mono* *<br>*Cornus controversa* *<br>*Fraxinus rhynchophylla*<br>*Juglans mandshurica*<br>*Quercus mongolica* **<br>etc. |
| **Understory tree layer** | *Lindera obtusiloba* ** | *Acer pseudosieboldianum* *<br>*Fraxinus rhynchophylla* *<br>*Lindera obtusiloba* **<br>etc. | *Lindera obtusiloba* **<br>*Magnolia sieboldii* *<br>etc. |
| **Shrub layer** | *Fraxinus sieboldiana* *<br>*Lespedeza bicolor* *<br>*Lespedeza cyrtobotrya* **<br>*Rhododendron micranthum* **<br>*Rhododendron mucronulatum* *<br>*Rhododendron schlippenbachii* **<br>*Toxicodendron trichocarpum* **<br>*Vaccinium hirtum* var. *koreanum* **<br>*Weigela florida* *<br>etc. | *Callicarpa japonica* *<br>*Clerodendrum trichotomum* *<br>*Fraxinus sieboldiana* **<br>*Lespedeza bicolor* *<br>*Lespedeza maximowiczii* *<br>*Lindera glauca* *<br>*Rhododendron mucronulatum* **<br>*Rhododendron schlippenbachii* **<br>*Rubus crataegifolius* *<br>*Toxicodendron trichocarpum* **<br>*Symplocos sawafutagi* *<br>*Vaccinium hirtum* var. *koreanum* **<br>etc. | *Alangium platanifolium* var. *trilobum* *<br>*Cimicifuga simplex*<br>*Corylus heterophylla*<br>*Rhododendron mucronulatum* **<br>*Styrax obassia* *<br>*Weigela subsessilis* *<br>etc. |

**Table 6.** *Cont.*

| Vegetation Stratum | Ridge | Slope | Valley |
|---|---|---|---|
| **Herb layer** | *Arundinella hirta* *** <br> *Athyrium yokoscense* ** <br> *Carex humilis* var. *nana* * <br> *Carex siderosticta* * <br> *Dendranthema zawadskii* var. *latilobum* * <br> *Disporum smilacinum* * <br> *Melampyrum roseum* * <br> *Miscanthus sinensis* var. *purpurascens* *** <br> *Pteridium aquilinum* var. *latiusculum* * <br> *Spodiopogon sibiricus* ** <br> etc. | *Ainsliaea acerifolia* * <br> *Artemisia keiskeana* * <br> *Aster scaber* * <br> *Athyrium vidalii* * <br> *Athyrium yokoscense* ** <br> *Atractylodes ovata* * <br> *Calamagrostis arundinacea* *** <br> *Carex humilis* var. *nana* * <br> *Carex siderosticta* * <br> *Disporum smilacinum* * <br> *Hydrangea serrata* f. *acuminata* <br> *Polystichum tripteron* * <br> *Potentilla freyniana* * <br> etc. | *Actinidia arguta* * <br> *Angelica decursiva* <br> *Athyrium yokoscense* ** <br> *Carex humilis* var. *nana* * <br> *Cimicifuga dahurica* * <br> *Corydalis speciose* * <br> *Dryopteris crassirhizoma* <br> *Isodon excisus* * <br> *Miscanthus sinensis* var. *purpurascens* ** <br> *Persicaria filiformis* <br> *Polystichum tripteron* <br> *Scutellaria dependens* * <br> etc. |

*, **, and *** indicate disappeared species, tolerant species, and pioneer species, respectively.

The restorative treatment was determined by reflecting the damaged level of vegetation and soil acidification. In very severely damaged zones, dolomite of 4.5 ton/ha and organic fertilizer of 2.25 ton/ha were recommended for soil amelioration. Plant species forming all layers of vegetation stratification, including canopy tree, understory tree, shrub, and herb layers, were recommended for vegetation restoration. In severely damaged zones, dolomite of 3.0 ton/ha and organic fertilizer of 1.5 ton/ha were recommended for soil amelioration. Plant species forming all layers of vegetation stratification were recommended for vegetation restoration, such as in the case of the very severely damaged zone. In moderately damaged zones, dolomite of 1.5 ton/ha and organic fertilizer of 0.75 ton/ha were recommended for soil amelioration. For restoring vegetation, plant species forming shrub and herb layers of vegetation stratification were recommended. Plant species forming shrub and herb layers were recommended in the moderately damaged zone. In the lightly damaged zone, dolomite of 1.0 ton/ha and organic fertilizer of 0.5 ton/ha were recommended for soil amelioration. Passive restoration was recommended for restoring vegetation.

In this restoration plan, we did not recommend for plant species forming the understory tree layer that vegetation to be introduced on the upper slope and ridge, as vegetation established there usually lacks the layer.

## 4. Discussion

### 4.1. The Effects of Air Pollution on Forest Ecosystems

Air pollution and atmospheric deposition emitted from industrial facilities have adverse effects on tree and forest health. Growing awareness of the air pollution effects on forests has, from the early 1980s on, led to intensive forest damage research and monitoring. This has fostered air pollution control, especially in developed countries, and also, to a smaller extent, in developing countries. At several forest sites, particularly in developed countries, there are the first indications of the recovery of forest soil and tree conditions that may be attributed to improved air quality [4,71]. This caused a decrease in the attention paid by the public to the air pollution effects on forests. However, air pollution continues to affect the structure and functioning of forest ecosystems just as when this study was conducted.

Air pollutants may impact trees as both wet and dry deposition. Wet deposition comprises rain, hail, and snow and is largely determined by atmospheric processes. Dry deposition consists of gases, aerosols, and dust and is largely influenced by the physical and chemical properties of the receptor surface. Forests receive higher deposition loads than open fields, depending on the tree species and canopy structure. A higher roughness of the canopy causes higher air turbulences and more intensive interactions between the

air and the foliage. The interception of pollutants by the foliage in turn is determined by such factors as leaf area index, leaf shape, leaf surface roughness, and stomata size. Dry deposition accumulated on the foliage is washed off by precipitation and enhances the deposition under the canopy (throughfall) in comparison to deposition in an open field (bulk deposition). Moreover, throughfall is influenced by two components of canopy exchange: canopy leaching and canopy uptake of elements. The main air pollutants involved in forest damage are sulfur compounds, nitrogen compounds, ozone, and heavy metals [72–76].

Sulfur dioxide ($SO_2$) was the first air pollutant found to cause damage to trees [77]. Its air concentrations increased rapidly when it was released into the atmosphere by the combustion of fossil fuels during the course of industrialization. While damaging trees directly via their foliage, $SO_2$ also reacts with water in the atmosphere to form sulfurous acid ($H_2SO_3$) and sulfuric acid ($H_2SO_4$), thus contributing to the formation of acid precipitation and hence to the indirect damage of trees [72,73]. In Korea, forest decline has usually occurred around industrial areas, and $SO_2$ has played a leading role [7,8,13]. However, Korea has shown declining trends in $SO_2$ concentrations in recent years [4].

Nitrogen oxides (NOx) are released into the atmosphere in the course of various combustion processes in which nitrogen (N) in the air is oxidized mainly to nitrogen monoxide (NO), with a small admixture of nitrogen dioxide ($NO_2$). In daylight, NO is easily converted to $NO_2$ by photochemical reactions involving hydrocarbons present in the air. Both gases, especially NO, are also produced biologically by soil bacteria during nitrification, denitrification, and decomposition of nitrite ($NO_2^-$) [78]. These substances are gaseous and act on trees as dry deposition directly via the foliage. Some of them are acidifying and lead—by means of chemical reactions with water in the atmosphere—to acid precipitation. Acidifying compounds such as $SO_2$, NOx, and $NH_3$, however, enhance the concentrations of protons and form sulfuric acid, nitric acid ($HNO_3$), ammonium ($NH_4$), and nitrate ($NO_3$) [73,74,79].

Heavy metals result from most combustion processes and from many industrial production processes. They are released into the atmosphere by means of dust and, at high temperatures, also as gases. The main heavy metals considered to be detrimental to forest health are cadmium (Cd), lead (Pb), mercury (Hg), cobalt (Co), chromium (Cr), copper (Cu), nickel (Ni), and zinc (Zn). However, largely because of their impacts on human health, heavy metal emissions have been reduced greatly within the last 3 decades in many industrialized countries [80–82].

### 4.2. Damage Status to Forest Ecosystem around Seokpo Smelter

Vegetation in places close to smelters has been so severely damaged that only a few plants, such as *Athyrium yokoscense* (FR. Et SAV.) H.CHRIST, *Miscanthus sinensis* var. *purpurascens* RENDLE, and *Pteridium aquilinum* var. *latiusculum* (DESV.) UNDERW, sporadically exist; otherwise all of them have disappeared. As the distance from the smelter become farther, the damage decreases, resulting in shrubland and forest (Figures 4 and 5). The spatial distribution of vegetation in industrial areas usually reflects the degree of pollution damage [4,7,8,12]. If a forested ecosystem is being affected by air pollution, then the canopy stratum is generally impacted first and is stripped away. As canopy trees decline, shrubs and then the ground vegetation are affected. This syndrome of the sequential death of the horizontal strata of the terrestrial vegetation, described as a "peeling" of "layered vegetation effect" by [62], was observed clearly in this area (Figure 5).

Forests appeared as two types depending on the damage degree. Forests with moderate damage show poor development of vegetation stratum, while forests with light damage show visible damage in appearance and poor development of undergrowth (Figure 5). The damage was also reflected in species composition (Figure 8) and species diversity (Figure 9), showing a clear difference in species composition and species diversity compared to the reference area.

The soil is acidified and has a lower calcium and magnesium content compared to the reference area, while the aluminum content was higher in very severely damaged sites (Figure 6).

Synthesized results obtained from the diagnostic assessments on vegetation damage and soil acidification in the forest ecosystem around the Seokpo zinc smelter show that vegetation damage was so severe that denuded ground appeared throughout wide areas, and soil acidification was also relatively severe. In this respect, active restoration is required urgently to prevent follow-up damage, such as landslides [83–85]. However, the damage decreased with increasing distance from the pollution source and was restricted within the first ridge from the source. The results of this diagnostic assessment could help to determine the spatial range and level of restoration.

In the case of the Yeocheon industrial complex, passive restoration occurred in forests around industrial complexes [4], but the speed was slower than in the case of the Ulsan industrial complex where active restoration was applied [8]. In active restoration, dolomite and sludge treatment neutralized acidic soil and supplemented nutrients, thereby facilitating plant growth and contributing to the restorative effects [8]. In addition, the tolerant species, which was selected through field surveys and laboratory experiments, was well established and contributed to achieve successful restoration [8,36].

*4.3. Necessity and Recommendation of Ecological Restoration*

In Korea, most industrial complexes are located on the coastal areas [4,7,8]. However, the area where this study was conducted is uniquely located on a small mountain village. Most of this area is composed of a mountainous area with a steep slope, and it is therefore not easy to develop. Therefore, the nature is well conserved. The mountainous land of this area is very steep and shallow in soil depth. Thus, Korean red pine forest, which is adapted well to the dry and infertile environment, dominates the vegetation of this area. However, deciduous broad-leaved forests suitable for the climate condition of this region and the environmental characteristics of each site are well developed in lowlands below midslope, including mountainous valleys. The river that runs through this area corresponds to the upstream section of the Nakdong River, one of the longest rivers in Korea. As Korea is a mountainous nation where more than 65% of the total national territory is composed of mountainous areas, most riparian zones, including floodplains of rivers, are transformed into agricultural lands and urbanized areas, leaving few integrate rivers equipped with aquatic and riparian zones in the plains and lands with a gentle slope. However, as this area is not easy to develop due to the environmental characteristics composed of steep mountainous areas; the river also remains an almost completely intact status.

However, air pollutants emitted from the Seokpo smelter have destroyed forest vegetation established in the basin of this river, even leading to landslides. Therefore, its impact poses a great threat to the river ecosystem conserved so well. Considering these facts comprehensively, the restoration of the damaged forest ecosystem is absolutely necessary and urgently required. In particular, countermeasures such as restoring forests are important for ensuring future ecosystem services [7,8,13].

The continual growth of the human population and human land uses leads to declines in the quality of the environment. Further, the natural landscapes that provide many ecosystem services are being rapidly converted to agricultural, industrial, and urban areas, and even wastelands. The biodiversity and habitability of the planet are now more threatened than ever before. Therefore, it is imperative that degraded land be rehabilitated and that adjoining natural landscapes be protected. However, it is clear that degradation thresholds have been crossed in many habitats, and succession alone cannot restore viable and desirable ecosystems without intervention [86]. As was shown in our results (Figures 4 and 5), the forest has degraded through shrubland or grassland to denuded land, eventually producing continual landslides. In addition, the soil is acidified and becoming non-nutritive (Figure 6). Thus restoration actions are urgently required to prevent more land degradation.

To restore degraded ecosystems, in particular those degraded by pollution, we need to apply soil ameliorators, including dolomite [6,8,13,87,88]. Although these soil amendments contributed to improving the polluted environment and thereby achieved successful revegetation in the case of Sudbury [6], they may cause other problems, such as ground water pollution and eutrophication [73,89,90]. In this respect, we recommend planting tolerant plants or applying fertilizer plants rather than applying soil amendments as a restorative treatment in all cases [8].

### 4.4. Soil Amelioration for Restoration

The atmospheric environment in the industrial areas in Korea is improving due to a decrease in the emission of air pollutants [4]. However, the polluted soil has not improved so easily [7,8]. In fact, polluted soil usually provides the major challenge to most restoration programs [8,13,29,35]. Therefore, soil amendment was planned as a preparation for restoration. The soil amelioration focused on the neutralization of acidic soil and the enhancement of fertility. Previous research showed that dolomite was a superior ameliorator compared to lime [15], and it is also generally used [8,16]. The amount of dolomite applied was calculated with an equation being applied to determine the amount of dolomite required to improve varying soil pH to 5.5 [15]. In the present situation, the amounts of dolomite required were 4.5, 3.0, 1.5, and 1.0 ton/ha in very severely, severely, moderately, and lightly damaged zones, respectively (Table 1). These amounts were smaller than those used in the Sudbury region of Canada [16] and the Ulsan industrial area of Korea [13].

Soil neutralized by the addition of dolomite stimulates activities of soil microorganisms and enhances nutrient availability through the promoted decomposition of organic matter [8,91]. Organic fertilizer also may ameliorate acidified soil by raising pH and adding macronutrients, including phosphorus, which is often a limiting nutrient in acidic soil [13,88,92].

Aluminum toxicity results in rapid inhibition of root growth due to the impedance of both cell division and elongation [93–95], reduction of soil volume explored by the root system, and direct interference with the uptake of ions such as calcium and phosphate across the cell membrane of damaged roots [96,97]. The deficiency in soil nutrients, such as P, $Ca^{2+}$, and $Mg^{2+}$, exacerbate the problem of inefficient nutrient uptake due to restricted root growth and root damage [98,99].

Additions of undecomposed plant materials, such as pruning, to acid soils often increase soil pH, decrease $Al^{3+}$ saturation, and improve conditions for plant growth generally [100–104]. Similarly, the addition of plant residue composts, urban waste compost, animal manure, and coal-derived organic products to acid soils increases soil pH, decreases $Al^{3+}$ saturation, and improves conditions for plant growth [8,13,105–107]. The recycling of these waste products for soil amelioration has a double benefit for both the environment and the economy, provided that the waste materials are not contaminated with harmful impurities. These organic substances confer metal binding and pH buffering capacities, which are important determinants of the pH of the treated soil [104,108,109].

Treatment of sludge as a soil ameliorator contributed to the reduction of $Al^{3+}$ content and resulted in increased plant growth [8]. This result suggests that the sludge is a chelating agent for $Al^{3+}$ [104,109,110]. Although dolomite and sludge contribute to ameliorating the acidified soil, there are some serious concerns for the land application of dolomite and sewage sludge due to the potential for the contamination of ground water and eutrophication [73,88,89,110]. By stimulating the mineralization of soil organic matter, dolomitic liming causes ground water pollution by increasing nitrate release from the soil [8,89,90]. We, therefore, recommend restricting the use of those soil ameliorators.

Meanwhile, N-fixing plants have been used for enhancing soil fertility in revegetation projects elsewhere (e.g., [6,13,16]). Therefore, we recommend planting N-fixing plants as an initial step for revegetating in this area.

*4.5. Selection of Plant Species for Restoration*

This study corresponds to a diagnostic assessment which analyzed the damage status of the forest ecosystem as a preparatory step for realizing the ecological restoration of this area [19,20]. Furthermore, we carried out a vegetation survey to obtain the reference information in a conservation reserve with similar environmental conditions, which was designated as the forest genetic resource reserve from the Korea Forest Administration.

Studies for restoration have chosen the species for restoration on the basis of the following criteria: (1) species importance for restoring the ecosystem function, (2) species that are to be the main components of the final ecosystem, and (3) many plants that make up the final biodiversity of the ecosystem and should be able to recolonize by their own efforts [7,8,13,35].

In this study, we first selected the species lost in this study area by comparing and analyzing the vegetation data obtained from the damaged and natural reference areas.

Next, we selected tolerant species which can withstand the polluted environment. Tolerant plants were selected as species flourishing specifically in the damaged area and species showing higher frequency in the damaged area than in the reference area. Tolerant plants were selected by classifying the four vegetation layers of canopy tree, understory tree, shrub, and herb composing the vegetation profile.

Finally, we selected the pioneer species frequently invading the bare ground to restore the heavily damaged zone. Synthesized, the plant species to be introduced for restoration were selected by combining the tolerant species with the polluted environment, the pioneer species frequently invading with the bare ground, and the species disappeared from the damaged area compared with the species composition of the reference area (Table 6) [7,13,36].

## 5. Conclusions

The result of a diagnostic assessment for the damaged forests around the Seokpo zinc smelter showed that ecological restoration is required urgently, as vegetation damage and soil acidification are very severe within the first ridge and at a distance of about 5 km from the pollution source. Vegetation damage appeared variously, such as the level to which bare ground appears due to extreme damage and the landslides that follow, the degree to which some of the vegetation strata is lost, and the visible damage level. The degree of soil acidification tended to be proportional to vegetation damage. In relation to soil acidification, deficiencies in nutrients, such as $Ca^{2+}$ and $Mg^{2+}$, and an increase in toxic ion concentration, such as $Al^{3+}$, were also identified. In particular, landslides continued around places where vegetation was severely destroyed, making ecological restoration urgent. The restoration plan was prepared by compiling the results of these diagnostic assessments and reference information collected from intact natural forests. The restoration plan was prepared in two directions: amelioration of acidified soil and vegetation restoration. In order to successfully complete this ecological restoration, the continuous monitoring and management of soil and air pollution need to be prepared, as well as the continuous monitoring of the establishment process of the vegetation that is to be introduced.

**Author Contributions:** Conceptualization, A.R.K., C.H.L. and C.S.L; methodology, A.R.K., C.H.L. and C.S.L; software, A.R.K. and C.H.L.; validation, B.S.L., J.S. and C.S.L.; formal analysis, A.R.K. and C.H.L.; investigation, A.R.K., B.S.L., J.S., C.H.L., W.S.L., Y.H.Y.. C.S.L.; resources, C.H.L. and C.S.L.; data curation, B.S.L and J.S.; writing—original draft preparation, A.R.K.; writing—review and editing, Y.H.Y. and C.S.L.; visualization, A.R.K., B.S.L.; supervision, C.S.L.; project administration, C.S.L.; funding acquisition, C.S.L. All authors have read and agreed to the published version of the manuscript.

**Funding:** This research received no external funding.

**Conflicts of Interest:** The authors declare no conflict of interest.

# Appendix A

**Table A1.** Chemical Properties of Organic Fertilizer Planned as a Soil Ameliorator.

| Environmental Factors | Content |
|---|---|
| Water content (%) | 48.34 |
| Organic matter (%) | 33.76 |
| Total Nitrogen (%) | 1.24 |
| Available Phosphorus (%) | 1.04 |
| Exchangeable Potassium (%) | 0.26 |
| C.E.C (cmol+/kg) | 35.0 |
| Sodium (%) | 0.57 |

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
