# Peer review of "Diagnostic Assessment and Restoration Plan for Damaged Forest around the Seokpo Zinc Smelter, Central Eastern Korea"

_forests, doi:10.3390/f12060663_

Round 1
Reviewer 1 Report
The study aimed to restore the forest ecosystem damaged by air pollution. The authors stated that the main objective was to prepare the restoration plan based on the surveys taken in the damaged forest ecosystem in the vicinity of the zinc smelter and the reference information obtained from the intact forest ecosystem. The study meets the aims of MDPI Forests and contains information about the restoration survey of contaminated soil.
The background and the description of the study site are quite well prepared and give the reader a solid introduction to the problem. Also, the method section does not raise any objections. The results are clearly described and contain information in well-prepared figures.
During reading the manuscript I did not found significant proofreading flaws. The text is well written, the language is also correct.
Below I listed major comments:
The first question that arose during reading the manuscript concerns the novelty of the study. There is no information about it. I'm afraid that the used methods (known well in the reclamation of soils) and the preparation of the study based only on the restoration plan may not be sufficient to consider the manuscript to be highly significant and novelty. In my opinion, only the area may be new (however, I do not know if the chosen area was studied earlier) - there is also no information why the authors decided to choose such study site. Such information should be indicated clearly at the end of the introduction section (some information I manage to read in the discussion section (lines 487-491).
My second doubt concerns the discussion section:
In the discussion, in the first subchapter, the authors give overall information about the effects of air pollution on the forest ecosystems, indicating the role of major air pollutants. In the second sub. which I find the most important, there is a slight discussion with the results obtained. In this sub. I would rather add some additional references to discuss the results:
- l. 397 - references about preventing landslides should be added,
- l. 384-385 - are there any other studies conducted earlier that indicate the influence of distance on forest damage? In my opinion, it should be also supplemented with additional references.
- Subchapter 4.5 is, almost in all, repeating the information from the method section... It is redundant. In exchange, a part of it may be moved to the method section to supplement it.
- lines 496-498 - are repeating the information from the introduction. There is no need to repeat the objective of the study. It was mentioned once. This only increases the number of pages in the manuscript...
- I also think it would be better to move subsection 4.6 together with tables 5 and 6 to the results section just below table 4. It will be more appropriate and it will increase the readability of the restoration plan.
Line 466 - patents? I think it is redundant.
Overall comment: The manuscript is quite well written. However, it needs to be revised taking into account the listed comments. To summarize, the authors should add more references into the discussion section to discuss the results. They also should provide the background where they will indicate the significance and novelty of the study. Without pointing it out, the manuscript is only a well-prepared restoration plan.
Author Response
Response to Reviewers’ comments
We sincerely appreciate the valuable comments from the editor and reviewers and tried to incorporate the comments into the revised manuscript.
Response to reviewer 1
The study aimed to restore the forest ecosystem damaged by air pollution. The authors stated that the main objective was to prepare the restoration plan based on the surveys taken in the damaged forest ecosystem in the vicinity of the zinc smelter and the reference information obtained from the intact forest ecosystem. The study meets the aims of MDPI Forests and contains information about the restoration survey of contaminated soil.
The background and the description of the study site are quite well prepared and give the reader a solid introduction to the problem. Also, the method section does not raise any objections. The results are clearly described and contain information in well-prepared figures.
During reading the manuscript I did not found significant proofreading flaws. The text is well written, the language is also correct.
Below I listed major comments:
The first question that arose during reading the manuscript concerns the novelty of the study. There is no information about it. I'm afraid that the used methods (known well in the reclamation of soils) and the preparation of the study based only on the restoration plan may not be sufficient to consider the manuscript to be highly significant and novelty. In my opinion, only the area may be new (however, I do not know if the chosen area was studied earlier) - there is also no information why the authors decided to choose such study site. Such information should be indicated clearly at the end of the introduction section (some information I manage to read in the discussion section (lines 487-491).
☞ We explained the background of our work in this area in Introduction section of this manuscript. Lines 47 – 64. However, we reinforced it at the end of the introduction by accepting the reviewer's comments.
My second doubt concerns the discussion section:
In the discussion, in the first subchapter, the authors give overall information about the effects of air pollution on the forest ecosystems, indicating the role of major air pollutants. In the second sub. which I find the most important, there is a slight discussion with the results obtained. In this sub. I would rather add some additional references to discuss the results:
- 397 - references about preventing landslides should be added,
☞ We revised our manuscript by accepting the reviewer's comments. Line 476.
- 384-385 - are there any other studies conducted earlier that indicate the influence of distance on forest damage? In my opinion, it should be also supplemented with additional references.
☞ We revised our manuscript by accepting the reviewer's comments. Lines 459-464.
- Subchapter 4.5 is, almost in all, repeating the information from the method section... It is redundant. In exchange, a part of it may be moved to the method section to supplement it.
☞ In this part we are discussing the process of preparing for vegetation restoration. Method section is referring to specific methods for selecting species for vegetation restoration. So, this part differs from the Method section. In view of this, we would like to retain this part.
- lines 496-498 - are repeating the information from the introduction. There is no need to repeat the objective of the study. It was mentioned once. This only increases the number of pages in the manuscript...
☞ We revised our manuscript by accepting the reviewer's comments. Line 575.
- I also think it would be better to move subsection 4.6 together with tables 5 and 6 to the results section just below table 4. It will be more appropriate and it will increase the readability of the restoration plan.
☞ We moved this part to Results section by accepting the reviewer's comments. Lines 386-407.
Line 466 - patents? I think it is redundant.
☞ No. They are not patents. We referred this part to explain the necessity of organic fertilizer as a soil ameliorator.
Overall comment: The manuscript is quite well written. However, it needs to be revised taking into account the listed comments. To summarize, the authors should add more references into the discussion section to discuss the results. They also should provide the background where they will indicate the significance and novelty of the study. Without pointing it out, the manuscript is only a well-prepared restoration plan.
☞ We revised our manuscript by accepting the reviewer's comments. Lines 410-586.

Reviewer 2 Report
Forests-1205128 reviewer remarks
General remarks: The paper reports on the severe impacts of industrial air pollution on soil and plant communities in Korea. The paper requires significant revision, however, to improve focus and clarity. For example, the introduction section requires significant revision and expansion to provide appropriate context for the study. Upon further review, some of the required information is already present in the text, just in less appropriate sections. Furthermore, the paper is conceptually split between a thorough survey of existing conditions related to pollution and development (and application?) of a restoration approach involving soil amelioration and planting of appropriate species. This is too much. This paper should be revised to focus only on the problem (effects of pollution on soils and forest community), and a second paper could be written to describe the restoration approach and evaluate its effectiveness.
Specific remarks:
Introduction:
- The introduction section is too general. Condense the treatment of ecological restoration into a concise paragraph—this is a concept with which readers of Forests will likely be familiar, so it does not require as long of an overview as currently written.
- Furthermore, this section would be improved by including context more specific to the case at hand. Review the literature more pertinent to localized air pollution and environmental degradation related to industrial activity, such as smelting, other factories, etc. This will provide a context more relevant for your work.
- Describe pollutants emitted as a result of the smelting operation, perhaps in the third paragraph, starting with “pollutants discharged beyond the limits of the buffering capacity...”
- Pg 2 line 65: What is the connection between energy use and zinc smelting?
Materials and Methods:
- Line 147: indicate parenthetically that vegetation survey field checks will be described in further detail later
- Figure 1: The locations of the factories should be noted using points or symbols (e.g., squares, circles); it is difficult to tell the specific factory location with the present format—is it located in the center of the text descriptor, at one end, etc.
- The study site description is thorough; however, in some cases, citations are necessary to support descriptions. For example, citations supporting the temperature inversion dynamic in this system and the dominant forest communities in this region would strengthen the paper.
- What were vegetation cover classes used to estimate cover as part of the vegetation surveys?
- Move reference forest description to the study site section
- How were vegetation survey plots selected? Randomly? The methods describe the total number of plots surveyed, and the dimensions of the plots, but it is unclear how plots were distributed across the landscape.
- Line 186: Soil chemical analyses are described in appropriate detail, but the soil sampling approach is not described. Were soils sampled using a corer/auger? A sampling spade? To what depth? Were they composited? Were they collected in a grid, or within vegetation survey quadrats?
- How were soil amendments applied? The methods describe rates of application, but also need to describe how application sites were identified, what area was designated for amelioration, how amendments were applied/incorporated, etc.
- At some point in the paper, describe how this plant can continue to emit pollution within the present regulatory framework. In the introduction section, it is suggested that factories of small scale away from public attention are presently emitting, but this is unsatisfactory. Why/how is this plant not regulated? Is there a gap/loophole in policy that permits this, or is the plant truly just off the radar?
Results:
- Figure 4 (and other figures): A wastewater treatment plant and areas impacted by forest fire appear on the maps, but do not appear in figure captions or otherwise described in text. If these are relevant, I recommend addressing them in the study site description. If not, I recommend leaving them off the figures.
- Figure 7: Clarify how only a single value is expressed for the reference site. There are 10 plots per community type—was there no variability among reference community types? Some indicator of variability should be included, such as SD.
Discussion:
- Much of the air pollution information in the discussion section should be moved to the introduction section, such as the paragraphs starting in lines 345, 357, and 364.
- Why are the soils near the smelter acidified and cation-depleted? Explicitly connect this observed relationship to soil acidification and base cation depletion associated with sulfur and nitrogen oxide pollution.
- Section 4.4--description of rates and other information about soil amendment application should be moved to methods.
Author Response
Response to Reviewers’ comments
We sincerely appreciate the valuable comments from the editor and reviewers and tried to incorporate the comments into the revised manuscript.
Response to reviewer 2
General remarks: The paper reports on the severe impacts of industrial air pollution on soil and plant communities in Korea. The paper requires significant revision, however, to improve focus and clarity. For example, the introduction section requires significant revision and expansion to provide appropriate context for the study. Upon further review, some of the required information is already present in the text, just in less appropriate sections. Furthermore, the paper is conceptually split between a thorough survey of existing conditions related to pollution and development (and application?) of a restoration approach involving soil amelioration and planting of appropriate species. This is too much. This paper should be revised to focus only on the problem (effects of pollution on soils and forest community), and a second paper could be written to describe the restoration approach and evaluate its effectiveness.
Specific remarks:
Introduction:
- The introduction section is too general. Condense the treatment of ecological restoration into a concise paragraph—this is a concept with which readers of Forests will likely be familiar, so it does not require as long of an overview as currently written.
☞ We revised Introduction section of our manuscript by accepting the reviewer's comments. Lines 96-138.
- Furthermore, this section would be improved by including context more specific to the case at hand. Review the literature more pertinent to localized air pollution and environmental degradation related to industrial activity, such as smelting, other factories, etc. This will provide a context more relevant for your work.
☞ We revised Introduction section by reflecting the specificity of the area and the level of ecological restoration of the developing countries including South Korea.
- Describe pollutants emitted as a result of the smelting operation, perhaps in the third paragraph, starting with “pollutants discharged beyond the limits of the buffering capacity...”
☞ We revised Introduction section of our manuscript by accepting the reviewer's comments. Lines 96-104
- Pg 2 line 65: What is the connection between energy use and zinc smelting?
☞ It means fossil fuel energy, which use in the process of refining Zinc. Seokpo smelters refine primarily processed ore rather than raw ore.
Materials and Methods:
- Line 147: indicate parenthetically that vegetation survey field checks will be described in further detail later
☞ Field check referred here means identification of vegetation patches.
- Figure 1: The locations of the factories should be noted using points or symbols (e.g., squares, circles); it is difficult to tell the specific factory location with the present format—is it located in the center of the text descriptor, at one end, etc.
☞ We revised Figure 1 by accepting the reviewer's comments.
- The study site description is thorough; however, in some cases, citations are necessary to support descriptions. For example, citations supporting the temperature inversion dynamic in this system and the dominant forest communities in this region would strengthen the paper.
☞ We revised our manuscript by accepting the reviewer's comments.
- What were vegetation cover classes used to estimate cover as part of the vegetation surveys?
☞ We applied Braun-Blanquet’s cover class and added it in our manuscript.
- Move reference forest description to the study site section
☞ We moved description for the reference site to the study site section by accepting reviewer's comment.
- How were vegetation survey plots selected? Randomly? The methods describe the total number of plots surveyed, and the dimensions of the plots, but it is unclear how plots were distributed across the landscape.
☞ We revised our manuscript by accepting the reviewer's comments.
- Line 186: Soil chemical analyses are described in appropriate detail, but the soil sampling approach is not described. Were soils sampled using a corer/auger? A sampling spade? To what depth? Were they composited? Were they collected in a grid, or within vegetation survey quadrats?
☞ We revised our manuscript by accepting the reviewer's comments.
- How were soil amendments applied? The methods describe rates of application, but also need to describe how application sites were identified, what area was designated for amelioration, how amendments were applied/incorporated, etc.
☞ It is a plan and not practiced yet. Our restoration plan was prepared depending on vegetation damage degree as degree of soil acidification was closely related to the vegetation damage class.
- At some point in the paper, describe how this plant can continue to emit pollution within the present regulatory framework. In the introduction section, it is suggested that factories of small scale away from public attention are presently emitting, but this is unsatisfactory. Why/how is this plant not regulated? Is there a gap/loophole in policy that permits this, or is the plant truly just off the radar?
☞ It means that it has been able to avoid regulation in the past because it is located so remote that it does not receive public attention. However, the pollution damage has been revealed and is now in the public's attention. It is therefore currently complying with emission standards. However, we did not feel the need to deal with this in my academic paper, so we’d like to omit it.
Results:
- Figure 4 (and other figures): A wastewater treatment plant and areas impacted by forest fire appear on the maps, but do not appear in figure captions or otherwise described in text. If these are relevant, I recommend addressing them in the study site description. If not, I recommend leaving them off the figures.
☞ We revised such figures by accepting the reviewer's comments.
- Figure 7: Clarify how only a single value is expressed for the reference site. There are 10 plots per community type—was there no variability among reference community types? Some indicator of variability should be included, such as SD.
☞ We revised such Figure 7 by accepting the reviewer's comments.
Discussion:
- Much of the air pollution information in the discussion section should be moved to the introduction section, such as the paragraphs starting in lines 345, 357, and 364.
☞ We supplemented Introduction section. So, we’d like to keep this part in Discussion section.
- Why are the soils near the smelter acidified and cation-depleted? Explicitly connect this observed relationship to soil acidification and base cation depletion associated with sulfur and nitrogen oxide pollution.
☞ Seokpo smelters use much fossil fuel energy in the operating process of smelter and emit pollutants. The pollutants acidified soil around the industrial facilities. The soil acidified as such contains low content of basic cations, Ca and Mg because those cations were exchanged with hydrogen ion, H+ and leached away. Our result reflect the trend (Figure 6).
- Section 4.4--description of rates and other information about soil amendment application should be moved to methods.
☞ We explained soil amelioration methods in Method section. So, we’d like to keep this part in Discussion section.

Round 2
Reviewer 1 Report
All of the included comments satisfied me, therefore I have no doubts that the paper may be accepted for publication.
Author Response
Response to Reviewers’ comments
We sincerely appreciate the valuable comments from the editor and reviewers and tried to incorporate the comments into the revised manuscript.
Response to reviewer 1
All of the included comments satisfied me, therefore I have no doubts that the paper may be accepted for publication.
☞ Thank you for your valuable comments.
Reviewer 2 Report
Forests-1205128_r1
Reviewer Comments: I thank the authors for their revision and clarifications in responses to my previous comments. I recommend further careful editing to ensure there is no confusion about implementation of the restoration plan--specific examples noted below.
Line 17: If the restoration plan was not put into practice as part of this study, it is incorrect/misleading to state that the study was conducted to “ecologically restore the forest ecosystem damaged by air pollution.” As written, and as I understand it, this study primarily characterized the degradation related to air pollution and then developed a restoration plan that could be used later.
Line 307: If the restoration plan were not yet implemented, it is unclear/misleading to say that “dolomite and organic fertilizer were applied...”
Lines 386 – 394: Fairly repetitive; the second and third paragraph here repeat what is said in the first paragraph. Revise to state these results clearly and concisely.
Lines 585-588: clarify that the active restoration mentioned here occurred at a different site; this contributes to confusion about whether or not the restoration plan were implemented.
Lines 622-623: as above, contributes to confusion about whether or not the restoration plan were implemented: “although those soil amendments contributed to improving the polluted environment...”
Figures: I see that points for factory locations were added to one figure, but I’m not seeing these points on the rest of the figures. Please update all maps with discrete points showing factory locations.
Author Response
Response to Reviewers’ comments
We sincerely appreciate the valuable comments from the editor and reviewers and tried to incorporate the comments into the revised manuscript.
Response to reviewer 2
Reviewer Comments: I thank the authors for their revision and clarifications in responses to my previous comments. I recommend further careful editing to ensure there is no confusion about implementation of the restoration plan--specific examples noted below.
Line 17: If the restoration plan was not put into practice as part of this study, it is incorrect/misleading to state that the study was conducted to “ecologically restore the forest ecosystem damaged by air pollution.” As written, and as I understand it, this study primarily characterized the degradation related to air pollution and then developed a restoration plan that could be used later.
☞ We revised our manuscript by accepting reviewer’s comment. Lines 17-18.
Line 307: If the restoration plan were not yet implemented, it is unclear/misleading to say that “dolomite and organic fertilizer were applied...”
☞ We revised our manuscript by accepting reviewer’s comment. Lines 271.
Lines 386 – 394: Fairly repetitive; the second and third paragraph here repeat what is said in the first paragraph. Revise to state these results clearly and concisely.
☞ We revised our manuscript by accepting reviewer’s comment. Lines 591-594.
Lines 585-588: clarify that the active restoration mentioned here occurred at a different site; this contributes to confusion about whether or not the restoration plan were implemented.
☞ We revised our manuscript by accepting reviewer’s comment. Lines 508-510.
Lines 622-623: as above, contributes to confusion about whether or not the restoration plan were implemented: “although those soil amendments contributed to improving the polluted environment...”
☞ We revised our manuscript by accepting reviewer’s comment. Lines 548-548.
Figures: I see that points for factory locations were added to one figure, but I’m not seeing these points on the rest of the figures. Please update all maps with discrete points showing factory locations.
☞ We revised Figures 3, 4, 5, and 6 by accepting reviewer’s comment.

This manuscript is a resubmission of an earlier submission. The following is a list of the peer review reports and author responses from that submission.
Round 1
Reviewer 1 Report
Air pollution and atmospheric deposition have adverse effects on tree and forest health. This study deals with Forest Decline and Restoration Plan of Degraded Forest under the Heavy Air Pollution around the Seokpo Zinc Smelter, Central Eastern Korea. The aim of this study is is to (1) analyse the present status of a forest in an ecological restoration context, (2) produce a restoration gap analysis of this forest based on ecological qualities of a reference state, and (3) discuss the results and implications for restoration.
However, the manuscript presented is more of a technical report than a scientific research paper, describing the development of a restoration plan.
Detailed comments:
- Introduction
L 50: In order to highlight the particular risks of soil erosion and landslides, references to surface relief, to geology and to the monsoon climate would have been important since rainfall is concentrated in a short monsoon period (peaks in July 295 mm and Aug. 242 mm) (Climate data: https://en.wikipedia.org/wiki/Bonghwa_County).
L92-98: instead of the clearly stated research questions (or hypotheses) in the Introduction section, there is a description of the working procedure.
- Material and methods
2.1 Study site
L 104: a relief map would have been helpful, and in addition, an overview table showing the percentage of classified slope inclination levels.
Information on climate (annual mean temperature, annual precipitation, precipitation seasonality, and other meaningful bioclimatic variables [https://www.worldclim.org/data/bioclim.html]), altitude above sea level, geological substrates etc. are completely absent.
L112: It is not transparent by whom and on what basis the vegetation types were defined. Do they result from a classification of your own data in combination with vegetation mapping in the field? Do they result from vegetation classification by remote sensing? Were they adopted from an existing vegetation map (or vegetation survey)? There is lack of transparency and replicability.
Fig. 1: it is impossible to identify the vegetation mapping units the in the map legend.
L121: the figure caption is incomplete and misunderstandable. What you mean by: “Colored map expressed vegetation types established the smelter”?
2.2 Methods
L146 and L154: What was the minimum distance between the plots in order to reduce bias of the spatial auto-correlation caused by sampling?
L147-155: What were we the exact criteria for the selection of the natural reference forests? To what extent were the accordance with the ecological basic conditions taken into account (e.g., surface relief, altitude above sea level, geological substrate, soil type, water balance, soil nutrient level, soil depth etc.)? Which deviations were allowed up to which level? Again, the scholary interest in transparency, reproducibility and replication are not sufficiently.
L153: What you mean by: “The reference forests were selected as the forests, which are from 50 to 100 year-old that can form a stable forest and avoid the old growth forest”?
Furthermore, it is unclear, why the existing data and results from the current work of Kim & Chon (2017) on “Vegetation Composition and Structure of Sogwang-ri Forest Genetic Resources Reserve in Uljin-gun, Korea” were not included in your data set. Here is an excerpt from the abstract and from the keywords line of Kim & Chon (2017):
“Based on a total of 272 vegetation data collected by the ZM school phytosociological study method, the composition and structural characteristics of the forest vegetation in the Sogwang-ri forest genetic resource reservoir located in Uljin-gun, Gyeongsangbuk-do were compared using the table comparison method and the TWINSPAN method, And their ecological characteristics were analyzed. The types of forest vegetation were classified into 7 types, and it was divided into two major groups, 'Slope and Ridge type', which characterized by Quercus mongolica, Pinus densiflora for. erecta, Lespedeza bicolor etc. and 'valley and concave slope', which characterized by Cornus controversa, Fraxinus mandshurica, Morus bombycis, Hydrangea serrata for. acuminata etc. The hierarchy of the vegetation unit was 2 community groups, 4 communities, and 6 subcommunities.
Keywords: …phytosociogical study; slope and ridge type; valley and concave type, potential natural vegetation”.
Is there scientifically sound reason for excluding this current and relevant reference? Or was is simply overlooked?
Kim, H.Y.; H.J. Cho. Vegetation Composition and Structure of Sogwang-ri Forest Genetic Resources Reserve in Uljin-gun, Korea. Korean J. Environ. Ecol. 31, 2017, 188-201, https://doi.org/10.13047/KJEE.2017.31.2.188
L156-167: I wonder why data on the concentrations of heavy metal content of the soil at the vicinity of the zinc smelter were omitted. For example, Li et al. (2015) found that the concentrations of smelter driven metals in topsoil decreased with increasing distance from the smelter. They found the main contamination by Pb, Zn, and Cd in the upper 40cm of soil around the Pb/Zn smelter, but traces of Pb, Zn, and Cd contamination were found below 100cm. Reference:
Li, P.; Lin, C.; Cheng, H.; Duan, X.; Lei, K. Contamination and health risks of soil heavy metals around a lead/zinc smelter in southwestern China. Ecotoxicol. Environ. Saf. 113, 2015, 391-399. doi: 10.1016/j.ecoenv.2014.12.025.
Is there scientifically sound reason for excluding the heavy metal concentrations from soil analysis?
- Results
3.2 Soil damage
The term `soil damage´ should be replaced by `soil degradation´ which is the “physical, chemical and biological decline in soil quality. It can be the loss of organic matter, decline in soil fertility, and structural condition, erosion, adverse changes in salinity, acidity or alkalinity, and the effects of toxic chemicals, pollutants or excessive flooding” (NSW Environment, Energy and Science).
3.3 Species composition
L228: misspelling error physic-chemical insteadt of physicochemical
L232-237: the figure caption is incomplete. Information on number of plots; number of species in total; number of plots around Seokpo smelter (n = ?) and in the natural reference forest, Uljin Forest Genetic Resources Conservation Reserve (n = ?); number of runs; number of iterations etc. is missing.
3.5 Zoning and design for restorative treatment
L249: misspelling error zonning instead of zoning
The Results chapter of a scientific research paper should represent the core findings of the study derived from the methods applied to gather and analyze information. However, subchapter 3.5 is not consistent with these principles. Some parts belong to the Methods section (L 257: “Dolomite requirement was calculated by applying the following equation…). Other parts concerning phase like “We decided that…” (L 259) or “were recommended for soil for soil melioration” (L226), etc. belong to the Discussion chapter or to the Conclusions chapter.
L291: The level of detail in Tab. 2 is rather unspecific and tentative. Here, I would have expected validated results by site-specific analysis (e.g., indicator species analysis) with statistical output and significance level for all species. For those species that would have been validated to be indicative I would have appreciated any detailed information on important species traits, dominance, successional status, contamination tolerance, etc.
- Discussion
The Discussion chapter should focus on explaining and evaluating what you found, showing how it relates to your literature review and research questions, and making an argument in support of your overall conclusion. However, there are sections, which are not consistent with these principles.
4.4 Soil melioration
L414: A technical description, that „dolomite raises soil pH and increases available Ca2+ and Mg2+ due the following chemical reactions in soil solution…“ is not a discussion and is the wrong place her.
4.5 Selection of plant species of restoration
This paragraph is rather descriptive since many relationships, e.g. between the toxic stress parameters and the plant species of the tree layers, the shrub and the herb layer, should have been tested by the appropriate statistics. Therefore, I rate large fractions of the discussion as speculative or vague.
In addition to this serious deficit, I would have expected a debate with numerous publications on the phenomenon of species adaptation and speciation processes on heavy metal soils (rapid evolutionary adaptation to heavy metal-polluted soils in plant species of different genera). Numerous publications are dedicated to the topic of soil contamination by hyperaccumulation of Zn and Cd. This is a hot topic for basic research in the fields of ecology, evolution and conservation science. These plants include obligate metallophytes (true metallophytes), which are often endemic to their native metalliferous sites, for example:
Singh, M.; Kumar, J.; Singh, S.; Singh, V.P.; Prasad, S.M.; Singh, M.P.V.V.B. Adaptation strategies of plants against heavy metal toxicity: A short review. Biochemical Pharmacology (Los Angel) 2015, 4, 161-167.
Emamverdian, A.; Ding, Y.; Mokhberdoran, F.; Xie, Y. Heavy Metal Stress and Some Mechanisms of Plant Defense Response. The Scientific World Journal 2015, Article ID 756120, pp. 18 https://doi.org/10.1155/2015/756120
Stevanović, V.; Tan, K.; Iatrou, G. Distribution of the endemic Balkan flora on serpentine I. - obligate serpentine endemics. Plant Syst. Evol. 242, 149–170 (2003). https://doi.org/10.1007/s00606-003-0044-8
Borymski, S.; Cycoń, M.; Beckmann, M.; Mur, L.A. J.; Piotrowska-Seget, Z. Plant Species and Heavy Metals Affect Biodiversity of Microbial Communities Associated With Metal-Tolerant Plants in Metalliferous Soils. Frontiers in Microbiology 2018, 9, Article ID 1425, https://doi.org/10.3389/fmicb.2018.01425
There are also natural serpentine soils with a potentially toxic concentration of heavy metals in Korea:
Kim, J.M.; Yang, K.C.; Choi, S.K.; Yeon, M.H.; Shin, J.H., Shim J.K. Plant uptake of heavy metals in Andong serpentine soil. Korean J. Environ. Biol. 2006, 24, 408-415
Ryou, S.H.; Kim, J.M.; Sim, J.K. Studies on the Decomposition of Leaf Litter Containing Heavy Metals in Andong Serpentine Area, Korea I. Microcosm Experiment. Korean J. Environ. Biol. 2009, 27, 353-362
It is unclear why this aspect was not considered, either for the research design or in the Discussion chapter.
- Conclusions
To me, the Conclusions chapter reads more like a summary. Summary refers to the concise statement or account of the key points of a research. The conclusion is that section of the text, which serves as the final answer to the research question. Hence, I come back to my comment on the Introduction chapter. Since no specific research questions were formulated at the beginning, there can be only a summary at the end.
However, at least I would have expected recommendations for establishment of an effective monitoring system. Monitoring air pollution is not only indispensable for identifying the cause of tree decline. It is also essential for measuring restoration success.
Reviewer 2 Report
Review of forests-1151628 “Forest decline and restoration plan of degraded forest under the heavy air pollution around the Seokpo zinc smelter, central eastern Korea”
General comments
This is a well written paper on an interesting topic of degradation and restoration after pollution damage to a forested ecosystem. The writing in English language is very well done but needs some editing for readability and to correct minor typos. Perhaps the title should include “case study” since there is no replicated research study. To me it appears to be a monitoring report that might be published as a case study after future re-measurements to assess change over time - and not original research - so I recommend reframing the paper as a review paper on smelter pollution damage and restoration literature, with a small case study added at the end. The Discussion lacks comparisons of findings with published studies which points to the problem of lack of results available for comparison.
Specific comments
Introduction
Consider adding a topic sentence to beginning of each paragraph and then linking them using the last sentence in each paragraph.
Methods
L140-142 Consider adding some detail to help explain this step.
The analysis of data as 5 categories is not ideal for a continuous variable such as percent “degradation” that could be analyzed using regression techniques.
Results
Figure 3 & 4 Consider reporting or studying how well the field data were predicted by the remote sensing approach (e.g., percent concordance)
Discussion
L295-338 this background material should probably be in Introduction section
L343+ try to avoid just restating the Results – repetitive
Conclusion
Good concise summary